# Convolutional Neural Networks or Vision Transformers: Who Will Win the Race for Action Recognitions in Visual Data?

**DOI:** 10.3390/s23020734

**Published:** 2023-01-09

**Authors:** Oumaima Moutik, Hiba Sekkat, Smail Tigani, Abdellah Chehri, Rachid Saadane, Taha Ait Tchakoucht, Anand Paul

**Affiliations:** 1Engineering Unit, Euromed Research Center, Euro-Mediterranean University, Fes 30030, Morocco; 2Department of Mathematics and Computer Science, Royal Military College of Canada, Kingston, ON 11 K7K 7B4, Canada; 3SIRC-LaGeS, Hassania School of Public Works, Casablanca 8108, Morocco; 4School of Computer Science and Engineering, Kyungpook National University, Daegu 41566, Republic of Korea

**Keywords:** convolutional neural networks, vision transformers, recurrent neural networks, conversational systems, action recognition, natural language understanding, action recognitions

## Abstract

Understanding actions in videos remains a significant challenge in computer vision, which has been the subject of several pieces of research in the last decades. Convolutional neural networks (CNN) are a significant component of this topic and play a crucial role in the renown of Deep Learning. Inspired by the human vision system, CNN has been applied to visual data exploitation and has solved various challenges in various computer vision tasks and video/image analysis, including action recognition (AR). However, not long ago, along with the achievement of the transformer in natural language processing (NLP), it began to set new trends in vision tasks, which has created a discussion around whether the Vision Transformer models (ViT) will replace CNN in action recognition in video clips. This paper conducts this trending topic in detail, the study of CNN and Transformer for Action Recognition separately and a comparative study of the accuracy-complexity trade-off. Finally, based on the performance analysis’s outcome, the question of whether CNN or Vision Transformers will win the race will be discussed.

## 1. Introduction

With the emergence of deep learning, computer vision (CV) pushed the limits of what was possible in the digital world [1,2,3]. Over recent years, problems that were assumed unsolvable are now being solved with super-human accuracy. The main reason for this success is the great diversity of the market and needs. New tasks such as medical imaging, Industry, Object Recognition [4], Autonomous Vehicle Navigation [5], Face Detection [6], Fingerprint Recognition [7], Fast Image Processing [8], and Robotic Navigation [9] have been tested at high accuracy. Furthermore, integrating artificial intelligence in image recognition is the subject of many uses.

NLP, or natural language processing, is a revolution in how computers and other technological devices are used. It is a processing system that translates human instructions into computer language and the other way around. As a result, the user interface is significantly more convenient and accessible. One of the most significant breakthroughs in the NLP field during 2022 has been creating machine learning models that create texts from scratch, with the GPT-3 (Generative Pre-Trained Transformer 3) [10] leading the way. The peculiarity of Transformers is that they can understand the context of words in a way that was not possible before. Remarkably, however, recent work demonstrated that Vision Transformers could also have equal or higher performance on large-scale image classification tasks [11].

Video understating like Action Recognition (AR), which we specifically deal with in this review, involves two famous techniques: Convolutional Neural Networks and Vision Transformers.

In point of fact, multiple different approaches to deep learning are still being utilized in several applications (Figure 1). Convolutional neural networks, on the other hand, have been shown to be the most successful model for dealing in computer vision with image and video data, and they are the ones that are used the most.

During this time, Transformer is the trendiest. In light of these considerations, the purpose of this paper is to conduct an investigation of both of these techniques on a more in-depth level.

Action recognition is core work in video understanding [12], which refers to recognizing an action in a video based on the complete execution of the action, reconciling the characteristics of video image data to achieve high-level understanding. It has been studied for decades, solving many problems such as abnormal event detection [13], video retrieval [14], intelligent robots [15], and visual surveillance [16].

The next Section describes the mechanism of CNN and its evolution over time, Section 2 represents some of CNN’s applications, and Action Recognition using CNN is summarized in Section 3. A description of Transformer models and architecture is given in Section 4. In the same way, as in Section 4, Action Recognition using this time transformer is given in Section 6. Section 7 compares these two models of the Deep Learning approach in terms of performance and complexity. Section 8 concludes the paper.

## 2. Convolutional Neural Networks, Review

Convolutional neural networks are inspired by the human visual system. The CNN model mimics the cortical area’s structure by going back to 1962. Hubel et al. [17] introduced the hierarchical model of the visual system based on an experimental study. The study shows that the cortex has very tiny areas of cells that are sensitive to specific parts of the field of vision, a primary area that specifically detects dark and bright spots, as well as the edges of the visual scene, and a secondary area that interprets the visual information.

Inspired by this discovery, in 1980, the first version of the CNN was reported by Fukushima [18], which introduced a neural network model for a visual recognition technology to recognize patterns named Neocognitron. The network is comprised of two layers of cells linked, similar to the biological human visual system. 

Stimulated by that, LeCun et al. [19], in 1989, gave a multi-layer network containing seven learned layers for handwriting digit recognition by applying a backpropagation method learning to handwrite digit images without the complex prepossessing stage. The architecture includes four convolutional layers, a pooling layer, and followed by three fully connected layers. The method revealed good results compared to the existing ones. However, as a consequence of the lack of trainable information and computing power, this architecture failed to work well under technical issues. 

After that, the same authors introduced a new class of neural architecture in 1998 named the LeNet [20]. One of the most common structures of this class is the LeNet-5. This class has seven layers of neural network architecture, without inputs. It is composed of two alternate convolutions and pooling layers followed by three fully connected layers at the end. Convolutions to hold the spatial direction of features, the downsampling of the feature space is performed by the average pooling along with a sigmoid activation between layers.

At the time, it was assumed that a better algorithm would always give better results regardless of the data. 

In 2012, Krizhevsky et al. [20] proved this theory wrong by announcing a deep CNN model named AlexNet for the image classification task. The model was record-breaking in the image recognition task. It successfully classifies 1.2 million high-res images from the Large-Scale Visual Recognition Challenge (LSVRC) challenge into 1000 different categories using purely supervised learning.

The most remarkable algorithm would only work correctly if the data learned represented the physical world—consequently, the project aimed to map the entire world of objects.

The architecture contains five convolutional layers, each one followed by max-pooling layers, and three fully connected layers with a final 1000-way Softmax. Due to the size of the network, the researchers applied data augmentation on the image data to expand the dataset using label-preserving transformations and dropout to reduce the overfitting problem. Figure 2 illustrates some of the famous architectures of CNN over time).

### 2.1. CNN Mechanism

CNN is a supervised deep learning model composed of a series of layers, each with specific functionality, which typically takes an image as input, then several hidden layers and an output.

CNN can handle complex images, unlike the multi-layer perceptron (MLP), which flattens image input, a 3 × 3 matrix of values separated by its three-color planes—red, green, and blue—into a vector and sends it to a multi-level perceptron. This causes a medium accuracy score when predicting classes but would have limited or no accuracy when it comes to complicated images with pixel dependencies everywhere or otherwise. 

ConvNets capture an image’s spatial and temporal dependencies, and the architecture better adapts to the image data set by minimizing the number of parameters implied and reusing the weights. Technically, the network is trained to understand image details better and scalable to massive data. 

The CNN method consists of applying different hidden layers, respectively. It generally has three major neural layers: convolution layers, pooling layers, and fully connected ones. Each type of layer performs a specific role and converts the input volume into an output neural activation volume. The utilization of a convolution layer on an RGB (red, green, and blue) image is depicted in Figure 3.

### 2.2. Convolution Layer

The major component of CNN and where most of the computation is involved. It is the layer that takes advantage of the inherent properties of the image and the layer to apply to the input image. Its parameters are centered on using kernels/filters/feature detectors.

These learning filters are relatively tiny relative to the size of the image in the spatial dimension. Therefore, the size determines the characteristics that the filters can select. Usually, these filters in the first layers can only extract features that cover up to 0.24% of the screen.

The filters are square matrices that map onto the image data by applying the dot product to the subregion of the input data and obtaining the output as a dot product matrix; that is to say, a 2D activation map. For greater accuracy in image analysis, it is recommended to add padding to the image, which is a margin of zero values placed on the edge of the image, and a practical technique to develop the depth so that the output of the current convolutional layer does not evolve small in dimension. 

As presented in GoogleNet, the convolution steps can be customized by specifying the number of pixels shifted on the input matrix. This number is in the stride. The following formula is used to determine the spatial dimension of the output of the convolution layers:(1)$$\frac{\left(V-K\right)+2Z}{S+1}$$
where V is the input size (Input height + padding height top + padding height bottom), K  represents the kernel size, Z is the quantity of zero padding set, and S is the stride.

### 2.3. Nonlinearity Layer

The nonlinearity Layer makes the precedent layer nonlinear by applying an activation function, like Rectified Linear Unit (ReLU), the Sigmoid function, or the hyperbolic tangent function in any of its layers or even in more than one layer. 

ReLU function is simple and fast and helps the training phase to converge reliably by outputting the input directly y is positive. Otherwise, it will output 0 (Equation (2)), while the **SOFTMAX** activation function is typically applied at the end of the final layer to convert the network’s output into a probability distribution. The formula is given in Equation (3):(2)$$\mathrm{Relu}\left(x\right)=f\left(x\right)=\{\begin{array}{c} 0,\;\;\mathrm{for}\;y<0\\ y,\;\;\;\;\mathrm{for}\;y>0 \end{array}$$
(3)$$\mathrm{Softmax}\left(z\right)=\frac{e^{z_{i}}}{\sum_{j=0}^{K}e^{z_{j}}}$$
where z represents the input vector, $e^{z_{i}}$ is the standard exponential function for the input, k is the number of classes and $e^{z_{j}}$ refers to the standard exponential function for the output.

### 2.4. Pooling Layer

This layer takes the convolution layer’s output and reduces the feature maps’ dimensions by summarizing the subregions, i.e., taking the common of the maximum value. The purpose of using this layer is to optimize the complexity of the model procedure and control the overfitting problem. The most popular pooling operation is max pooling, which involves taking the maximal value of each sub-region to reduce the dimensional scale. 

The max-pooling layer is a 2 × 2 kernel dimensional with a stride of 2 on the spatial size of the input while keeping 25% of the original dimension and the depth volume at its initial size. Another well-known pooling layer, average pooling, consists of calculating the average of each sub-region of the activation map from any 2 × 2 square in the activation map.

### 2.5. Fully Connected Layer

The objective of this layer is to flatter all the high-level features learned by all the convolution layers and mix all the elements for learning the non-linear combinations of the features. The layer is a feed-forward neural network that forms the last layer of the CNN network. Each neuron has connections to all neurons in the previous layer, and its connection has its weight. The activation is calculated by a matrix operation followed by a bias gap.

## 3. CNN-Related Work

In the past seven years, numerous studies and applications based on CNN have been developed in various domains, including healthcare [26] and autonomous vehicles [27].

Based on Web of Science databases, Figure 4 depicts the number of publications per year that mention CNN. Object detection is one of these applications, with the goal of identifying the object with a bounding box and determining its classification.

One-stage processes, such as YOLO [28,29], SSD [30], and CornerNet [31,32], can be distinguished from two-stage techniques, such as R-CNN [33], Fast R-CNN [34], and Faster R-CNN [35], which made a breakthrough in object detection.

In the two-stage object detection process, region proposals are picked beforehand, and then CNN classifies the items. In a single step, the model simultaneously returns the category probability and position coordinates of the objects.

CNN [36], a biometric identification approach based on characteristics of the human face, is currently employed in the real world for face detection, a significant application.

Deepface [37] and DeepID [38] performed better than humans in unconstrained circumstances for the first time using the LFW dataset [39].

Detection, alignment, extraction, and classification constitute the DeepFace procedure. After detecting the face as the input to the CNN, Taigman et al. [37] trained the model using the Siamese network, achieving state-of-the-art performance. While DeepID directly enters two face pictures into CNN to extract feature vectors for classification, CNN is used by DeepID to extract feature vectors. Recent advancements in face recognition have primarily concentrated on the loss function.

FaceNet [38], which was proposed by Google in 2015, employs a 22-layer CNN to train a model with 200 million photos, including eight million humans. FaceNet substitutes softmax with triplet loss to discover more effective embeddings.

## 4. CNN for Action Recognition

CNN’s models are now dominating action recognition (AR) in video and visual data. The objective of AR is to extract spatial information and motion over time, making video processing more complex than images. However, understanding human actions remains challenging due to the lack of equity concerning performance evaluation related to datasets, backbone choices, and experimental techniques [40].

Unlike image recognition, where ImageNet [41] has been the ideal benchmark for evaluation, the Kinetics dataset [42] is now the most popular reference for action recognition. Although, the kinetics dataset is highly biased in favor of spatial modeling, which ill-judged the validation of a spatiotemporal modeling capability model.

CNN-based action recognition generally offers three ways: 2D convolution [43,44,45] or 3D convolution [46,47,48] or both [49,50]. However, 2D and 3D methods are different in terms of feature extraction. Hence in the case of 2D convolution, a feature map extracts only spatial (two-dimensional) information, which involves adding another model to capture temporal information (fusions). In contrast, 3D convolution methods offer both spatial and temporal knowledge for a set of continuous images simultaneously. 

Figure 5 illustrates the difference between 2D convolutions and 3D convolutions processing. The question is how they perform in contrast to each other concerning the Spatiotemporal modeling of video data. The 3D CNN is an extension of the success of 2D models in image recognition [51] to recognize actions in videos. 

The objective is to extract spatiotemporal features directly from several video frames by applying a 3D filter on several adjacent video frames. Thus, motion information is caught. The operation at the position in the feature map in the layer is formalized as follows:(4)$$v_{il}^{xy}=\emptyset \left(b_{i,j}+\underset{m}{\sum }\overset{P_{i}-1}{\underset{p=0}{\sum }}\overset{Q_{i}-1}{\underset{q=0}{\sum }}\overset{R_{i}-1}{\underset{r=0}{\sum }}W_{i,j,m}^{p,q,r}.\;\;V_{i-1,\;\;\;m}^{x+p,\;\;\;y+q,\;\;\;z+r}\right)$$
where *ϕ* is the non-linear activation function, and w is the 3D weight matrix. *P*, *Q*, and Rare, respectively, the filter’s height, width, and temporal length. One of the essential works on 3D architectures for action recognition dates back to 2012. The authors designed a 3D model for action recognition for the first time.

The model consists of a wired layer that generates the gray, gradient, and optical flow channels, followed by 3D convolution and subsampling applied to each channel and finally producing the final action by aggregating the information from all channels. 

In 2015, Tran et al. presented a study [52] to find the most appropriate temporal length for 3D CNN and developed a VGG-style 3D CNN architecture. The same authors performed a search for 3D CNN in a deep residual learning framework and employed a ResNet18 style 3D CNN architecture named Res3D; this surpasses C3D by a considerable margin in terms of accuracy and recognition time. 

The 2D approach is reached with two-stream models [53], processing the RGB images and the optical flow separately in two CNN models with late fusion in the higher layers [54].

In that respect, a 2D CNN model for image-level feature extraction and an additional model for temporal information capture. For example, the TRN [55] method relies on many features to structure images’ relationships. The TSM [45] approach shifts some channels across the temporal dimension, allowing the exchange of information between neighboring images, or the TAM [40] method, which relies on 1 × 1 convolution in depth to capture temporal dependencies between images efficiently.

Various methods of temporal aggregation of feature descriptors have also been proposed. There are also more complex approaches that have been studied on how to capture the long term. These models have achieved SOTA performance on multiple large-scale benchmarks for instance, Kinetics [42], Something-Something [56], Sports1M [54], and Youtube-8M [57]. However, there is no winner between these approaches; 3D models perform more sufficiently than 2D standards on the Kinetics dataset, while 2D methods perform better on Something-Something.

## 5. Vision Transformer

In the beginning, transformer models [58] were only applied to Natural Language Processing (NLP). The transformers were used in text classification [59], language translation [60], and question answering [61]. 

For example, Vaswani et al. [62] developed a Transformer technique that was built on paying attention to activities related to machine translation. In 2008, a research group from Google led by Devin et al. [63] introduced BERT (Bidirectional Encoder Representations from Transformers), which consisted of 340 million parameters that had a huge impact on the future. On the basis of the BERT mechanism of self-attention, the obtained results have powerful representational capabilities that enable them to extract intrinsic characteristics.

The model pre-trains a Transformer on an unlabeled text that considers each word’s context (it is bidirectional). The recent mixture of transformer approaches can reach a huge 1.6 trillion parameters and contains multiple FPNs [64]. The Transformer is achieving the SOTA performance in different NLP datasets such as Glue [65], SQuAD 2.0 [66], and Swag [66].

All this proves that Transformer has already dominated in NLP applications by showing better performance and speed than RNNs models, thus raising the question of the possibility of leading the computer vision community and overtaking CNN. 

By all accounts, Transformer is already contributing to the computer vision domain (Figure 6), showing excellent results in different applications. For instance, object detection [67], segmentation [68], video understanding [51], and the like, as well as achieving the top performance on different image recognition benchmarks.

### 5.1. The standard Vision Transformer 

We present in this sub-section the general components of a Vision Transformer [11]. It is based typically on two major elements: a linear projection of an image and an encoder transformer that contains several MLP neural network models and a self-attention mechanism.

#### 5.1.1. Patch Embedding

The standard method of Vision Transformer consists of partitioning the input image into separate patches of the same shape as a sequence of embedded words used when applying transformers to natural language. In other words, the Visual Transformer splits the image into visual tokens $\left(x_{1},x_{2},\;\ldots x_{n}\right)$ by $X\;\in \;R^{n\times d}$. While CNN uses pixel arrays, it is required to specify the patch size n. This stage consists of flattening the image patches returned, which means the vectorization of the patches into vectors, projecting the flattened patches into a lower-dimensional space by applying the linear function to each vector $X_{n}$. 

The output is $Z_{n}=W_{xn}+b$ referred to as the patch embeddings. While W and  b are two shared parameters learned from the training data, they also add a position embedding learned from patches p ∈ 1, 2, …, n to the Z vectors so that the Z vector captures content and position at the same time [11,69]. As a result, closer patches tend to have comparable similarity position embedding than others. 

In classification tasks, another point to consider is adding another embedding learnable vector $z_{0}$ to the sequence X, which is the CLS token, to accumulate and store information extracted from the other tokens with the same shape as the other Z vectors [11].

#### 5.1.2. Transformer Encoder

At this level, the approach consists of applying the encoder transformer [59]. The Multi-Head Self Attention layer is the major component of this procedure, applied to the sequence $z_{1},z_{2},\;\ldots$ besides the MLP model, Layer Norm (LN) is integrated before each block (Figure 7), and a residual connection is added to Multi-Head Attention. 

That is to say, if the major element of CNN is convolution, then Self-attention is the major element of the Transformer. The self-Attention layer captures long-term dependencies between all inputs and aims at transforming one element into another, in contrast to the short memory of RNN models that usually forget the content of the distant position and mixes the contents of adjacent positions. It receives n entities without context and outputs n entities with contextual information. 

In other words, the Self-attention layer takes the inputs in the form of $\left(x_{1},x_{2},\;\ldots x_{n}\right)$ by $X\;\in \;R^{n\times d}$, and applies three learnable weight matrices (Queries $W^{Q}\in \;R^{d}\;*\;d_{q}$, Keys $W^{k}\in \;d\;*\;d_{k}$ and Values $W^{V}\in \;d\;*\;d_{k}$). 

Technically, comparing the query with all keys, re-weight, and aggregating the values with weights. The output of the self-attention layer is in the formula below:(5)$$\mathrm{Attention}\left(Q,K,V\right)=z=\mathrm{softmax}\left(\frac{Q.K^{T}}{\sqrt{d_{q}}}\right)V$$
where $z\;\in \;R^{n\times d}$ and **softmax** to get the attention score, with $Q\;={\;\mathrm{XW}}^{Q}$, $K\;={\;\mathrm{XW}}^{K}$ and $V\;={\;\mathrm{XW}}^{V}$ and the computation is the dot product.

Vision transformer uses Multi-Head Self Attention instead of a Self-Attention Layer, where the number of heads is generally eight for longer-term dependencies and to compress multiple complex relationships between different elements in the sequence, that is, the combined independent multiple self-attention that have the same input and do not share parameters $W_{i}^{Q}$, $W_{i}^{K}$, $W_{i}^{V}$ where i=0,…(h−1) and h is the number of the attention blocks, $W_{i}^{Q}\in \;R^{d\times d_{k}}\;$, $W_{i}^{K}\in \;R^{d\times d_{k}}\;$ and $W_{i}^{V}\in \;R^{d\times d_{k}}\;$.
(6)$$\mathrm{MultiHead}\left(Q,K,V\right)=Concat\left(head_{1},\ldots ,head_{h}\right)\;W$$
where $head_{i}=Attention\left(QW_{i}^{Q}\;,KW_{i}^{K}\;,\;VW_{i}^{V}\;\right)$, the results are then concatenated into one matrix $\;\left[C_{0},C_{1},\;\ldots C_{h-1}\right]\;\in R^{h.d\times d_{k}}$

#### 5.1.3. Pure Transformer

Image processing is more challenging than text processing, given the high dimensions, noise, and redundant modality. The researchers proposed several innovative architectures in a very short time to address these challenges, such as positional coding and normalization strategy. 

The first Vision Transformer approach was purely named ViT, realized by Google Research and Brain Team [11] for the classification task applied directly to image patches, as explained in the previous section. 

ViT generally needs to be pre-trained on large datasets and fine-tuned to slighter tasks. If not, when trained in medium-sized datasets, it produces weak accuracy, such as the ImageNet dataset. The authors state that it is advantageous to use higher resolutions in the fine-tuning part than in the pre-training part.

Although ViT can capture long-range dependencies between patches, it does not consider regional feature extraction because the 2D patch is projected onto a vector with a single linear layer. Many works have proposed to solve the problem of localizing visual information. 

For instance, TNT [70] splits the original patch into several sub-patches, and it introduces a new architecture, “Transformer in Transformer”, which uses an internal transformer block to map the relationship of the sub-patches and an external transformer block for the exchange of information at the patch location. 

Chu et al. [71] introduced a method, “Twins”, which is a shifted window-splitting approach for cross-window connections to perform alternately local and global attention layer by layer. 

Afterward, Z. Huang et al. introduced another method called Shuffle [72], which consists of using the spatial shuffle operation instead of the staggered window partitioning to allow cross-window connections. In contrast, the method of RegionViT [73] generates both regional and local tokens from the image. Hence, the local tokens get global information via attention to the regional tokens. T2T presents the aggregation of local features to enhance local information [74].

On the other hand, many approaches have been proposed to enhance the computation of self-attention, such as DeepViT [75], which proposed a method to increase diversity at different layers through inter-head communication to regenerate attention maps. 

KVT [76] implements K-NN attention to use only the locality of image patches and compute only attention with the most similar tokens. At the same time, XCIT [77] performs self-attention computation on feature channels instead of tokens for effectively processing high-resolution images.

## 6. Transformers for Action Recognition

Action Recognition (AR) for Vision Transformers is a suitable target. In the same way as language modeling, in which words or input characters are represented as a set of tokens, videos are defined as a set of successive frames [51]. The Transformer encoder not only makes training and inference more powerful by not involving costly 3D convolutions, but also allows a complete video to be processed in a single pass.

Although CNN is still the most widely used, it is limited concerning long-term dependencies, either in space or time. Thereby, long-range dependencies can only be captured when these operations are repeatedly applied during the repetition of local operations has several limitations, is computationally inefficient, and creates optimization problems that must be handled carefully. 

Neimark et al. [51] introduced a method named VTN, which discarded the standard approach of video action recognition based on 3D ConvNets, and set up a transformer-based method that takes into account all the information in the video clip, applying first the SOTA of 2D architectures to learn spatial features, add temporal information in the data stream using attention mechanisms on the resulting features by only inputting the RGB frames. 

The temporal part of the VTN method follows the Longformer method presented in [78], which was addressed first for text processing to handle long sequences. Longformer deals with all the tokens in the input sequences with a within-reach complexity (O(n)) using sliding window attention. In contrast, the BERT classification token [63] is given via a fully connected layer to identify events or actions. 

It showed competitive results in terms of accuracy, while the training and running were 16.1 and 5.1 faster during inference compared to the SOTA with different backbones.

Girdhar et al. [79] proposed an Action Transformer approach that gathers all human-specific contextual cues in the video clip to capture only the semantic context of others’ actions.

For instance, focus on hands and face, two essential elements to identify an action. The method stands on two networks, the base, and the head. In the base network architecture, a 3D CNN architecture is applied to produce features transmitted to the Region Proposal Network (RPN) to get object proposals. 

The Action Transformer Head applies self-attention layers on the person Box as a query (Q), while the features from the neighboring video clip are used as key (K) and value (V). 

The self-attention layer is applied to add the context of other present people and objects to the query vector to facilitate subsequent classification. The key and the value features are calculated as a linear projection of the original feature map from the base network and are tensors of size 16 × 25 × 25 × 128, while the query is a 128-dimensional vector. Following the Transformer encoder mechanism (Equation (6)), the operation is represented as:(7)$$Z^{\left(r\right)}=\underset{x,y,t}{\sum }softmax\left(\frac{Q^{r}K_{x}^{T}}{\sqrt{D}}\right)V_{xyt}$$

Q(r) corresponds to the features extracted by RPN by applying the scalar product on K features, normalized by $\sqrt{D}$. The resulting query is of the following form (Equation (8)). 

The authors in [80] utilized a dropout to $Z^{r}$ and appended it to the original query feature after it is passed through a residual branch consisting of a LayerNorm operation, followed by a Feed-Forward Network (FFN). It is implemented as a 2-layer MLP and a dropout. The final feature is passed across another LayerNorm to get the updated query ($Q^{\prime \prime }$).
$$Q{\left(r\right)}^{\prime }=\mathrm{LayerNorm}\left(Q\left(r\right)\right)+Dropout\left(A\left(r\right)\right)$$
(8)$$Q{\left(r\right)}^{\prime \prime }=\mathrm{LayerNorm}\left(Q{\left(r\right)}^{\prime }\right)+Dropout\left(FPN\left(Q{\left(r\right)}^{\prime }\right)\right)$$

The authors used only RGB data as input and obtained good results at the semantic level. Additional modalities, such as motion/flow, are likely to improve efficiency and, therefore, increase computational cost.

Arnab et al. [81] proposed multiple pure transformer models for video classification called ViVit. They considered two methods for video embedding: Uniform frame sampling, which follows the mechanism, and Tubelet embedding, which is an extension of ViT embedding that corresponds to 3D convolution. 

The results of the embedding are passed to the sequence of the video V ∈ RT × H × W × C to obtain a set of tokens which are the input of the transformer $z\;\in \;R^{n_{t}\times h_{t}\times w_{t}\times d}$. The authors proposed several variants that factorize the Spatiotemporal dimensions for the long token sequences encountered in videos. 

The first model presents a straightforward extension of ViT that forwards all Spatio-temporal tokens through the encoder; each transformer layer takes all pairwise interactions between each Spatiotemporal token, which poses the problem of complexity $\left(O\left(n^{2}\right)\right).\;$

The second model called a factorized encoder involves two separate transforming encoders, a spatial/temporal encoder; the first is used to model tokens extracted from the same temporal cue, and the second is used to model interactions between tokens from different temporal lines, while the output token from this encoder is ultimately classified (requires fewer FLOPs).

The third model, called Factorized Self-Attention, differs from the first model in that instead of calculating self-attention on all token pairs, a factorization aims to calculate self-attention first spatially and secondary temporally. Hence, each block of self-attention of the transformer modelizes the Spatiotemporal interactions. This model is better than model 1 in terms of efficiency and has the same complexity as Model 2. 

The last model, named Factorized dot-product attention, with the same computational complexity as Models 2 and 3 while keeping the same number of parameters as the non-factored Model 1, consists of factorizing the multi-head dot-product attention operation, i.e., modifying the keys and values of each query to look only at tokens of the same spatial or temporal index by building $K_{s}$, $V_{s}\in \;R^{n_{h}.n_{w}\times d}$ and $K_{t},\;V_{t}\in \;R^{n_{t}\times d}$, while half of the head’s attention is on the token spatial dimension $Y_{s}$, and the other half is for the token’s temporal dimension $Y_{t}$ , then merge the output of several heads by concatenating them and applying the linear projection $Y=concat\left(Y_{s}\;,{\;y}_{T}\;t\right).$

Plizzari et al. [82] suggested a two-stream Transformer network, as shown in Figure 8. On the one hand, A spatial-Self Attention (SSA) stream is applied in each frame to extract low-level features for embedding the relations between body joints [83], according to the self-attention formula, in a given frame at time t, for each node of the skeleton *i^t^*, a query vector $q_{t}^{i}\;\in \;R^{dq}$, a key vector $k_{t}^{i}\;\in \;R^{dk}$ and a value vector $v_{i}^{t}\;\in \;R^{dv}$ are obtained by applying trainable linear transformations to the features of the node $n_{i}^{t}\;\in \;R^{C_{in}}$ , shared by all nodes, of parameters $W_{q}\;\in \;R^{C_{in}\times d_{q}}$ , $W_{k}\;\in \;R^{C_{in}\times d_{k}}$ and $W_{v}\;\in \;R^{C_{in}\times d_{v}}$. 

Afterward, for each pair of body nodes $\;\left(i^{t},j^{t}\right)$, they applied a query-dot scalar product to get a weight $\alpha_{ij}^{t}\in R$
which denotes the power of the correlations among the two nodes. The outcome’s grade $\alpha_{ij}^{t}$ is used to weigh each joint value $v_{i}^{t}$ and a weighted sum is calculated to get a new embedding for the node $i^{t}$. 

The spatial formula of this approach is like the following:(9)$$\alpha_{ij}^{t}=q_{i}^{t}\times k_{j}^{t},z_{i}^{t}=\underset{i}{\sum }softmax\left(\frac{\alpha_{ij}^{t}}{\sqrt{d_{k}}}\right)v_{j}^{t}$$
where $W_{q}\in R^{Cha}$ (Cha is the number of output channels that form the new embedding of node $i^{t}$). The authors applied multi-head attention, which means repeating this operation H times, every single time with a distinct set of learnable parameters $\left(z_{i}1^{t},\;\ldots ,\;z_{i}H^{t}\right)$ referring to the same node being merged using a learnable transformation (Equation (7)) to get the output of SSA. 

On the other hand, to capture the change in some joints over time, they proposed a temporal self-attention stream (TSA), in which the dynamics of each joint are studied separately with all images. The formula is the same as (Equation (9)), where $i^{v}$, $j^{v}$ indicate the same articulation at different times. 

Figure 8 depicts spatial/temporal attention for the representation of skeletal data. The NTU-RGB+D 60 [84] and NTURGB+D 120 [85] datasets produced SOTA results for both models. They merged the two streams. In addition, they used a 2s-STTR architecture, similar to those presented in [86,87].

## 7. Comparative Study between CNN, Vision Transformer, and Hybrid Models

To ensure a fair comparison, we use a multi-criteria analogy term in this section. We first present the advantages and disadvantages of the considered approaches, and then outline the state of the art of each method in terms of their modality, regarding the most widely used academic action recognition datasets.

### 7.1. Pros and cons of CNN and Transformer

One of the most significant advantages of CNN is the sparse connectivity that is provided by the Convolutional layer. This sparse connectivity saves memory because it only needs a small number of weights and connections [52], which makes it memory effective. Because there are no weights allocated between two neurons of neighboring layers in the CNN, and the set of weights operates with one and all pixels of the input matrix [52], weight sharing is another essential feature of the convolutional layer. This feature helps reduce the time needed for training and the costs associated with training.

In their evaluation of previous work, Koot et al. [88] discovered that CNN performs better than Transformer when it comes to latency accuracy on lightweight datasets. CNN is also described to capture inductive bias, which is also known as prior knowledge, such as translation equivariance and localization while having pooling operation give partial scale invariance [89]. 

CNN, on the other hand, has a few weaknesses, including a slowness that is brought on by the max pooling operation, and, in contrast to Transformer, it does not take into consideration the several perspectives that can be gained by learning [90], which leads to a disregard for global knowledge.

Due to the fact that it offers solutions to CNN’s numerous weaknesses, the Transformer has quickly become CNN’s most formidable opponent. The capability of Transformer to prioritize relevant content while minimizing the repetition of content that is not important is one of the program’s capabilities [89]. In addition, because less demand is placed on the processing power, the visual characteristics are dynamically re-weighted based on the context [90]. This ability to mimic long-distance interactions in space and time, an essential requirement for visual movies [91], is another reason the Transformer stands out in front of CNN.

The transformer effectively encodes temporal data, an additional crucial component for action recognition. Lastly, multi-head attention, which is the booster part of the performance of vanilla self-attention and the essential component of the visual transformer process, affords the opportunity to learn many representations from various perspectives [92].

In spite of this, Dosovitskiy et al. [11] found in their research that transformer has a significantly lower level of image-specific inductive bias compared to CNNs. In order to overcome inductive bias, the model needs to be trained on very large and sufficient datasets so that it can figure out these image-specific features on its own based on the training examples. Therefore, it is important for the self-attention mechanism to automatically recognize the temporal relationships between video frames by searching through a huge database of video sequences. This is done in order to fulfill the requirements of the self-attention mechanism. The end consequence is longer training timelines, a significant increase in the demands placed on computer resources, and enormous datasets that need to be processed [90].

In light of what has been said, researchers are progressively merging these two models into a single model in order to leverage the complementary strengths of the two models and offset the flaws of the two models. 

The findings of Zhao et al. [93] demonstrate that Transformer and CNN are mutually supportive of one another and could both be integrated into a single predictive model. They developed a hybrid model by employing multi-stage convolutional layers as the backbone of the model and exchanging a few particular layers for transformer layers. This offers the hybrid model with the global and local modelling capabilities of transformers and convolutions, respectively.

Dosovitskiy et al. [11] also acknowledged it. Experiments were undertaken to demonstrate that the transformer model excels after training on the CNN model. In light of CNN’s ability to add location data to the Transformer model, it is important to note that the network is still in existence. Thus, a substantial amount of effort is required to add convolution to a typical transformer block. Hybrid models are approaches that mix CNN and Transformer. 

Table 1 depicts the complementarity of the two models. A summary of the most pertinent works is offered in the subsequent section.

Table 2 presents a comparison of the accuracy and complexity of the CNN, Transformer, and Hybrid model techniques for the recognition of actions.

The accuracy and complexity of CNN, Transformer, and Hybrid model techniques for Action identification on the UCF101 (UCF), HMDB51 (HDM), and Kinetics-400 (Kin) datasets are compared in Table 2 [106,107,108]. The datasets [UCF] [106], [HMDB51 (HDM) [107], and [Kinetics-400 (Kin) [108] are [UCF] [106], [HMDB51 (HDM) [107], and [Kinetics-400 (FLops).

### 7.2. Modality Comparative Literature

Visual data may originate from a range of modalities and may be utilised singly or in combination to describe a visual activity [109]. This review concentrates on the RGB data and the skeletal data. RGB data are commonly employed in real-world application scenarios since they are simple to acquire and provide a wealth of information [109].

Using the following RGB datasets, we evaluate the efficacy of the techniques under consideration: UCF101 [106], which contains 27 h of video data, over 13,000 clips, and 101 action classes, containing video data totaling over 13,000 min.

HMDB51 [107] consists of 51 action categories and 7000 manually annotated footage extracted from various sources, including digital movies and YouTube.

Kinetics-400 [42] includes 400 classes of human motion with at least 400 video clips for each class. Each clip lasts about 10 s. The accuracy metric is used to evaluate the models since all classes are similarly important.

In support of what has already been stated, Table 2 illustrates how performance and complexity vary from one model to the next by highlighting a variety of methodologies utilized for each RGB-based model.

In the Kinetics dataset, Xiong et al. [101] achieved the highest level of accuracy with 89.1% within the Google Research lab, exceeding the findings that have been considered to be state-of-the-art thus far. This in no way negates the fact that all the models discussed produce findings that are fairly encouraging, independent of the datasets and metrics that are taken into consideration. Each model tries to fix a distinct issue by considering the existing issues that have been found.

For example, Bonan Li et al. [105], based on a CNN architecture, addressed the problem of time-length movies with two attention mechanisms. This confirms the hybrid model’s viability as a method because it demonstrates that the problem can be solved using multiple attention mechanisms. Skeleton data are an additional acceptable modality for action recognition because they encode the trajectories of human body joints. These trajectories reflect meaningful human movements, simplicity, and informative representation, which are the primary characteristics of skeleton data.

The NTU-RGB+D 60 [84] and the NTU-RGB+D 120 [85] datasets are the most well-known examples of 3D action recognition. NTU-RGB+D 6O is comprised of 56,880 video clips and 60 activities that were carried out by 40 different individuals. Each individual human skeleton is made up of 25 joints, each of which has a unique set of three-dimensional coordinates. The NTU-RGB+D 120 database is an expansion of the NTU-RGB+D 60 database, and it contains 114,480 skeletal sequences of 120 action classes performed by 106 unique subjects. Cross-Subject Evaluation (CS) and Cross-View Evaluation are the metrics that are utilized in order to evaluate the reported outcomes in relation to these two benchmarks (CV).

CS evaluates the model according to the subjects who participated in the data set, while CV analyses the model according to the camera view. The evaluation results are presented for each of them as the classification accuracy is expressed as a percentage. CNN [110] served as the inspiration for the creation of a graph convolutional network (GCN), which was proposed to capture the structural relations among the data.

Because skeleton data occur naturally in graphs, numerous GCN techniques have been developed to represent skeleton data as graph structures consisting of edges and nodes. This is possible because skeleton data occur naturally in the form of graphs. Since GCN uses convolutions, works that employ this technology have been placed in the same area as CNN.

Table 3 is quite similar to Table 2. Still, instead of comparing models using the same datasets and metrics as Table 2, it compares models using various datasets and metrics described earlier in this section. Recent research has shown the effectiveness of Transformer and self-attention in resolving the same challenge as skeleton-based action recognition. This is despite the fact that CNN and GCN have made significant strides in solving this problem.

Qin et al. [111] used the same graph model that was used in GCN; however, they added a technique that gave the Transformer model with both the joint and bone representations of the skeleton as a single input.

**Table 3 sensors-23-00734-t003:** Skeleton modality comparison of CNN, Transformer, and Hybrid model techniques for action recognition on the datasets NTU-RGB+D 60 [84] and NTU-RGB+D 120 [85] in terms of the accuracy according to the Cross-Subject Evaluation (CS) and Cross-View Evaluation (CV).

	Model	Year	The Idea of the Model	Datasets (Accuracy)			
				NTU RGB+D		NTU RGB+D 120	
				CS	CV	CS	CV
**CNN**	Yan et al. [112]	2018	The authors developed the first strategy for collecting spatial and temporal data from skeleton data by encoding the skeleton data with GCN.	85.7	92.7		
	Banerjee et al. [113]	2020	The author developed a CNN-based classifier for each feature vector, combining Choquet fuzzy integral, Kullback–Leibler, and Jensen–Shannon divergences to verify that the feature vectors are complementary.	84.2	89.7	74.8	76.9
	Chen et al. [114]	2021	The authors employed a GCN-based method to model dynamic channel-by-channel topologies employing a refining technique.	93.0	97.1	89.8	91.2
	Chi et al. [115]	2022	The authors developed a novel method that combines a learning objective and an encoding strategy. A learning objective based on the information bottleneck instructs the model to acquire informative, yet condensed, latent representations. To provide discriminative information, a multi-modal representation of the skeleton based on the relative positions of joints, an attention-based graph convolution that captures the context-dependent underlying topology of human activity and complementing spatial information for joints.	92.1	96.1	88.7	88.9
	Song et al. [116]	2022	The authors developed a strategy based on a collection of GCN baselines to synchronously extend the width and depth of the model in order to extract discriminative features from all skeletal joints using a minimal number of trainable parameters.	92.1	96.1	88.7	88.9
**Transformer**	Shi et al. [117]	2021	The authors designed a pure transformer model for peripheral platforms or real-time applications. Segmented linear self-attention module that records temporal correlations of dynamic joint motions and sparse self-attention module that performs sparse matrix multiplications to record spatial correlations between human skeletal joints.	83.4	84.2	78.3	78.5
	Plizzari et al. [82]	2021	The authors proposed a novel method to the modelling challenge posed by joint dependencies. A spatial self-attention (SSA) module is used to comprehend intra-frame interactions between various body parts, while a temporal self-attention (TSA) module is used to describe inter-frame correlations.	89.9	96.1	81.9	84.1
	Helei et al. [80]	2022	The authors propose two different modules. The first module records the interaction between multiple joints in consecutive frames, while the second module integrates the characteristics of various sub-action segments to record information about multiple joints between frames.	92.3	96.5	88.3	89.2
**Hybrid models**	Wang et al. [111]	2021	The authors investigated employing a Transformer method to decrease the noise caused by operating joints. They suggested simultaneously encoding joint and body part interactions.	92.3	96.4	88.4	89.7
	Qin et al. [118]	2022	The authors proposed a strategy for concatenating the representation of joints and bones to the input layer using a single flow network in order to lower the computational cost.	90.5	96.1	85.7	86.8

## 8. Conclusions

This work examines CNN and Transformer for Action Recognition individually, as well as the trade-off between accuracy and complexity. In addition, this paper evaluates the majority of pertinent research emphasizing the benefits of each of the aforementioned tactics and their corresponding outcomes.

The challenge of visual action recognition is fraught with obstacles and limits. Since the quality of research has improved over time, it is evident that solutions are on the horizon for addressing these issues, whether by employing CNN or Transformer approach. Transformer, which is fairly new to the field of computer vision, has been quite competitive with CNN, which is ten years more established up to this point.

As for the primary question, and in light of this study, it should be mentioned that although both algorithms (i.e., CNN and Transformers) work in their way and have their own shortcomings and benefits, it is still difficult to determine who will win this race. Nevertheless, the hybrid method that is more efficient and cost-effective. It combines CNN with transformers to provide a reliable model. After all, the old adage asserts that working together is the key to success!

This hybrid model is the most attractive formula because it enables us to take advantage of a model’s strengths while simultaneously reducing the effects of that model’s downsides. Additionally, it has been demonstrated that hybrid models are highly useful for bridging the gaps generated by the deficiencies of specific models.

Therefore, we believe that this hybrid model might win the race. Furthermore, we anticipate a greater emphasis on testing this hybrid approach in action recognitions in visual data.

## Figures and Tables

**Figure 1 sensors-23-00734-f001:**
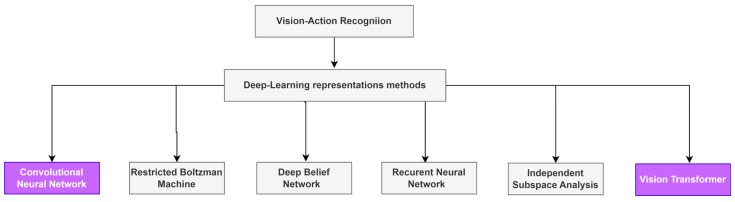
Different Deep Learning methods for Action Recognition.

**Figure 2 sensors-23-00734-f002:**
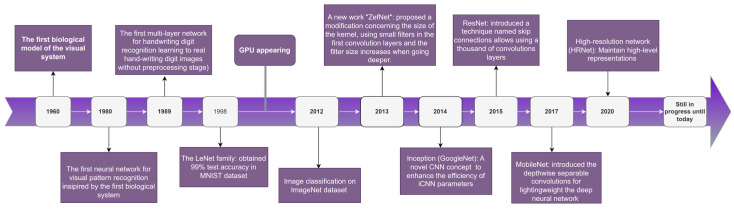
The evolution of CNN over time (Inception [21], ZefNet [22], ResNet [23], MobileNet [24], and HRNet [25]).

**Figure 3 sensors-23-00734-f003:**
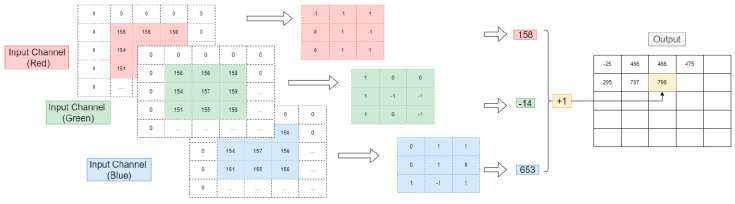
The application of convolution layer on RGB image.

**Figure 4 sensors-23-00734-f004:**
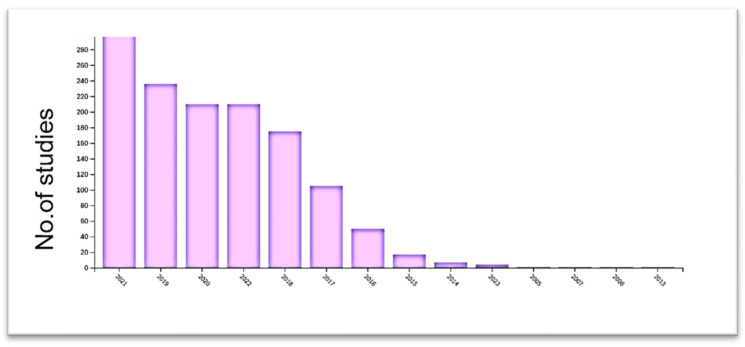
The distribution of CNN techniques in WOS on an annual basis. Since 2015, the number of studies on CNN applications in WOS has increased significantly.

**Figure 5 sensors-23-00734-f005:**
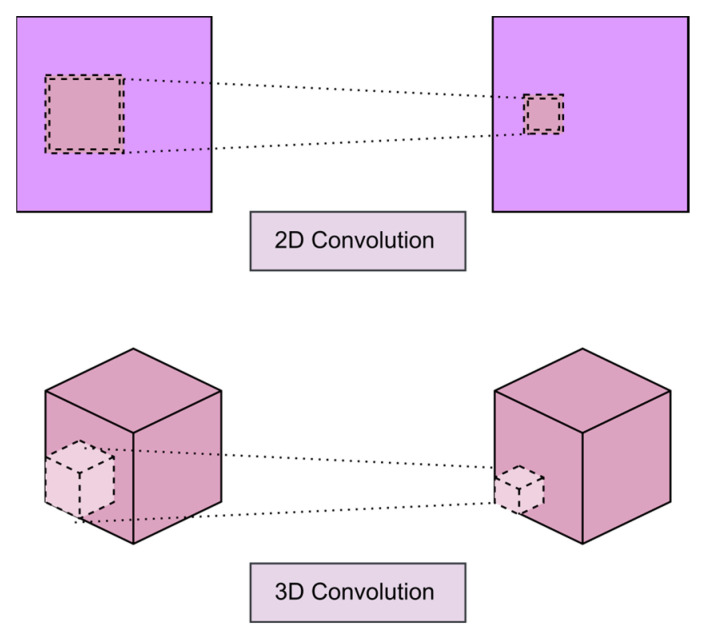
2D vs. 3D convolution layer.

**Figure 6 sensors-23-00734-f006:**
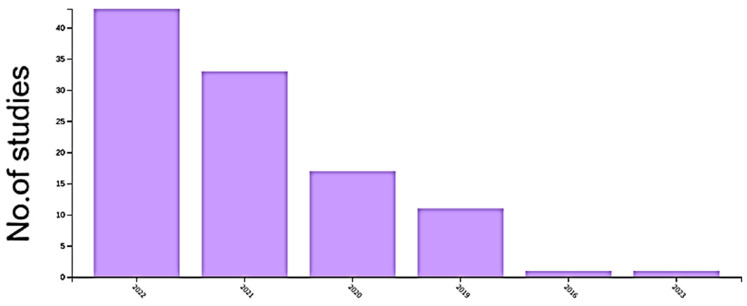
The Web-of-Science annual distribution of Vision Transformer techniques. Since 2020, the number of studies on Vision Transformer applications in WOS has expanded considerably.

**Figure 7 sensors-23-00734-f007:**
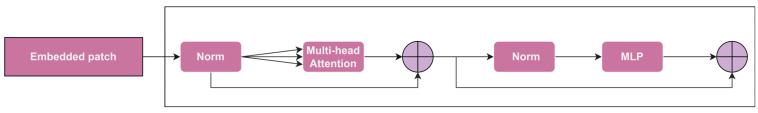
Transformer Encoder.

**Figure 8 sensors-23-00734-f008:**
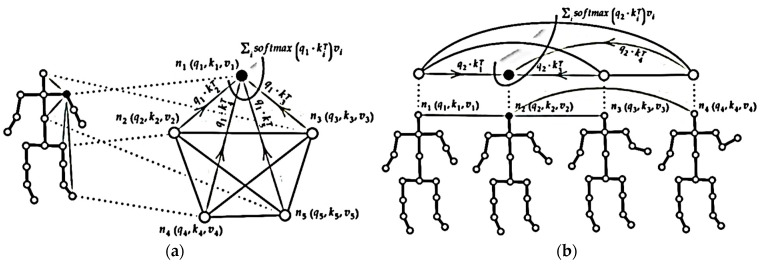
Two-stream Transformer network: (**a**) Spatial Self attention; and (**b**) Temporal self-attention Modified [82].

**Table 1 sensors-23-00734-t001:** CNN, Transformer, and hybrid model advantages, “-“indicates that the property is invalid in the method.

Properties	CNN	Transformer	Hybrid Model
Sparse connectivity	✓	-	✓
Weight sharing	✓	-	✓
Best at Latency accuracy on small datasets	✓	-	✓
Inductive bias	✓	-	✓
Capture local information	✓	-	✓
Dynamic weight	-	✓	✓
Capture global information	-	✓	✓
Learn from different angles	-	✓	✓
Best at Spatio-temporal Model long-distance interactions	-	✓	✓

**Table 2 sensors-23-00734-t002:** RGB comparison of CNN, Transformer, and Hybrid model approaches for Action recognition.

	Model + (Citation)	The Idea of the Model	Parameters	Flops	Year	Datasets (Accuracy)		
						UCF	HDM	Kin
**CNN**	Omi et al. [94]	Present a 3D CNN multi-domain-learning using adapters between layers. The results showed with ResNet backbone.	183.34		2022	63.33	93.29	67.80
	TEINet [95]	A technique for temporal modeling that enhances motion-related properties and adjusts the temporal contextual information channel-wise (backbone ResNet-50).		0.06	2020	96.7	72.1	76.2
	Xinyu Li [96]	Introduced a 3D CNN network that learns video clip-level temporal features from different spatial and temporal scales.	103	0.12	2020	97.9	75.2	
	SlowFast Networks [48]	A single-stream design that operates at two separate frame rates. SlowPath captures spatial semantics, but FastPath combines temporal semantics via the side connection. We displayed the outcomes using a 3D Resnet backbone.	32.88	0.36	2019			75.6
	Du Tran et al. [97]	Suggested factorizing 3D convolutions by separating channel interactions and spatiotemporal interactions in order to obtain greater precision at a lower computing cost.	32.8	1.08	2019			82.5
**Transformer**	Jue Wang et al. [98]	Dynamically predicts a subset of video patches to attend for each query location based on motion information.	73.9M	1.28	2022			79.0
	Arnab et al. [81]	Presented multiple models which factorize different components of the spatial-temporal transformer encoder. A solution to regularize the transformer model during training small datasets.		4.77	2021			84.9
	Liu et al. [99]	A pure transformer backbone model addressed the inductive bias of locality by utilizing the advantage of the intrinsic spatiotemporal locality in videos.			2022			84.9
	Yang et al. [100]	Fix the issue with videos’ needed set length. A strategy that uses the attention date to repeatedly construct interaction between the current frame input and the prior hidden state.	107.7	0.11	2022			81.5
	Xiong et al. [101]	A multi-view transformer is composed of multiple individuals, each of which focuses on a certain depiction. Through a lateral link between individual encoders, information from several visual representations is successfully merged.			2022			89.1
	Zha et al. [102]	The components of the Shifted Chunk Transformer are a frame encoder and a clip encoder. The frame encoder uses the picture chunk and the shifting multi-head self-attention elements to capture intra-frame representation and inter-frame motion, respectively.	59.89	0.34	2021	98.7	84.6	83.0
	Zhang et al. [103]	Using the proposed standard deviation, an approach that aggregates Spatial-temporal data with stacking attention and an attention-pooling strategy to reduce processing costs.	0.392	0.39	2021	96.7	74.7	80.5
**Hybrid- Model**	Kalfaoglu et al. [104]	Combining 3D convolution with late temporal modeling is a great way to improve the performance of 3D convolution designs.	94.74	0.07	2020	98.69	85.10	
	Bonan Li et al. [105]	The issue of time-length videos was solved by implementing two attention modules: a short-term attention module and a long-term attention module, each of which provided a distinct temporal token attention.		2.17	2022			81.3

## Data Availability

Not applicable.

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
