# Peer review of "Convolutional Neural Networks or Vision Transformers: Who Will Win the Race for Action Recognitions in Visual Data?"

_sensors, 2023, doi:10.3390/s23020734_

Round 1

Reviewer 1 Report

The manuscript provides a detailed description of the research and application of convolutional neural networks and Transformer-based models in the field of computer vision and action recognition. Especially for the application of Transformer in video action recognition, the author summarizes several mainstream frameworks for transformer application by comparing and analyzing the methods of video embedding and the processing of spatial and temporal features of the model. At the end of the paper, the author summarizes and compares the two methods of cnn and transformer, and points out the shortcomings of the transformer method in a more comprehensive way. For future research, this paper also puts forward two directions: in-depth study of transformer and fusion of cnn and transformer, which are more objective and comprehensive.

However, the content of this paper still has the following shortcomings:

1.     At the end of Section 3, the paper simply describes the performance of 2D and 3D cnn approaches on different datasets, and concludes that "However, there is no winner between these approaches" without analyzing the reasons in detail.

2.     The article describes in Section 6.3 that "despite the good results of Vision Transformer, the community still focuses on CNN structure. The article does not carry out an in-depth analysis on this point.

Author Response

REVIEWER #1

The manuscript provides a detailed description of the research and application of convolutional neural networks and Transformer-based models in the field of computer vision and action recognition. Especially for the application of Transformer in video action recognition, the author summarizes several mainstream frameworks for transformer application by comparing and analyzing the methods of video embedding and the processing of spatial and temporal features of the model. At the end of the paper, the author summarizes and compares the two methods of cnn and transformer, and points out the shortcomings of the transformer method in a more comprehensive way. For future research, this paper also puts forward two directions: in-depth study of transformer and fusion of cnn and transformer, which are more objective and comprehensive.

Dear reviewer, thank you for the time you took to study our work. After reading your constructive comments here are the modifications brought to the paper for each point.

Comment 1:   At the end of Section 3, the paper simply describes the performance of 2D and 3D cnn approaches on different datasets, and concludes that "However, there is no winner between these approaches" without analyzing the reasons in detail.

Author's response: We thank the reviewer for this comment. We have made the necessary changes to improve the results of our work.

Author's actions: Indeed, the comparison was kind of superficial, we rectified this important point by adding two sections named "Pros and cons of CNN and Transformer”, and “Modality comparative literature”, respectively. These two sections 7.1 and 7.2 in the new manuscript replaced the old sections 6.1 and 6.2. Section 7.1 enhances the particularity of each method and section 7.2 presents a thorough comparison of recent works in the literature of each method.

  • First, we updated all the figures, included the figure 5.
  • We added the following section and table.
  • We updated the conclusion (by answering the question who win the race.

To ensure a fair comparison, we use a multi-criteria analogy term in this section. We first present the advantages and disadvantages of the considered approaches, and then outline the state of the art of each method in terms of their modality, regarding the most widely used academic action recognition datasets.

7.1. Pros and cons of CNN and Transformer

One of the most significant advantages of CNN is the sparse connectivity that is provided by the Convolutional layer. This sparse connectivity saves memory because it only needs a small number of weights and connections [52], which makes it memory-effective. Because there are no weights allocated between two neurons of neighboring layers in the CNN, and the set of weights operates with one and all pixels of the input matrix [52], weight sharing is another essential feature of the convolutional layer. This feature helps reduce the time needed for training and the costs associated with training.

In their evaluation of previous work, Koot et al. [88] discovered that CNN performs better than Transformer when it comes to latency accuracy on lightweight datasets. CNN is also described to capture inductive bias, which is also known as prior knowledge, such as translation equivariance and localization while having pooling operation give partial scale invariance [89]. CNN, on the other hand, has a few weaknesses, including a slowness that is brought on by the max pooling operation, and, in contrast to Transformer, it does not take into consideration the several perspectives that can be gained by learning [90], which leads to a disregard for global knowledge.

Due to the fact that it offers solutions to CNN's numerous weaknesses, the Transformer has quickly become CNN's most formidable opponent. The capability of Transformer to prioritize relevant content while minimizing the repetition of content that is not important is one of the program's capabilities [89]. In addition, because less demand is placed on the processing power, the visual characteristics are dynamically re-weighted based on the context [90]. This ability to mimic long-distance interactions in space and time, an essential requirement for visual movies [91], is another reason the Transformer stands out in front of CNN.

The transformer effectively encodes temporal data, an additional crucial component for action recognition. Lastly, multi-head attention, which is the booster part of the performance of vanilla self-attention and the essential component of the visual transformer process, affords the opportunity to learn many representations from various perspectives [92].

In spite of this, Dosovitskiy et al. [11] found in their research that transformer has a significantly lower level of image-specific inductive bias compared to CNNs. In order to overcome inductive bias, the model needs to be trained on very large and sufficient datasets so that it can figure out these image-specific features on its own based on the training examples. Therefore, it is important for the self-attention mechanism to automatically recognize the temporal relationships between video frames by searching through a huge database of video sequences. This is done in order to fulfill the requirements of the self-attention mechanism. The end consequence is longer training timelines, a significant increase in the demands placed on computer resources, and enormous datasets that need to be processed [90].

In light of what has been said, researchers are progressively merging these two models into a single model in order to leverage the complementary strengths of the two models and offset the flaws of the two models. The findings of Zhao et al. [93] demonstrate that Transformer and CNN are mutually supportive of one another and could both be integrated into a single predictive model. They developed a hybrid model by employing multi-stage convolutional layers as the backbone of the model and exchanging a few particular layers for transformer layers. This offers the hybrid model with the global and local modelling capabilities of transformers and convolutions, respectively.

Dosovitskiy et al. [11] also acknowledged it. Experiments were undertaken to demonstrate that the transformer model excels after training on the CNN model. In light of CNN's ability to add location data to the Transformer model, it is important to note that the network is still in existence. Thus, a substantial amount of effort is required to add convolution to a typical transformer block. Hybrid models are approaches that mix CNN and Transformer.

Table 1 depicts the complementarity of the two models. A summary of the most pertinent works is offered in the subsequent section.

 Properties

CNN

Transformer

Hybrid model

Sparse connectivity

-

Weight sharing

-

Best at Latency accuracy on small
datasets

-

Inductive bias

-

Capture local information

-

Dynamic weight

-

Capture global information

-

Learn from different angles

-

Best at Spatio-temporal Model long-distance interactions

-

Table 1: CNN, Transformer, and hybrid model advantages,  “-“ indicates that the property is  invalid in the method.

Model + (Citation)

The idea of the model

Parameters

Flops

Year

Datasets (Accuracy)

UCF

HDM

Kin

CNN

Omi et al.

[94]

Present a 3D CNN multi-domain-learning using adapters between layers. The results showed with ResNet backbone.

183.34

2022

63.33

93.29

67.80

TEINet

[95]

A technique for temporal modeling that enhances motion-related properties and adjusts the temporal contextual information channel-wise (backbone ResNet-50).

0.06

2020

96.7

72.1

76.2

Xinyu Li

[96]

Introduced a 3D CNN network that learns video clip-level temporal features from different spatial and temporal scales.

103

0.12

2020

97.9

75.2

SlowFast Networks

[48]

A single-stream design that operates at two separate frame rates. SlowPath captures spatial semantics, but FastPath combines temporal semantics via the side connection. We displayed the outcomes using a 3D Resnet backbone.

32.88

0.36

2019

75.6

Du Tran et al.

[97]

Suggested factorizing 3D convolutions by separating channel interactions and spatiotemporal interactions in order to obtain greater precision at a lower computing cost.

32.8

1.08

2019

82.5

Transformer

Jue Wang1 et al. [98]

Dynamically predicts a subset of video patches to attend for each query location based on motion information.

73.9M

1.28

2022

79.0

Arnab et al.

[81]

Presented multiple models which factorize different components of the spatial-temporal transformer encoder. A solution to regularize the transformer model during training small datasets.

4.77

2021

84.9

Liu et al.

[99]

A pure transformer backbone model addressed the inductive bias of locality by utilizing the advantage of the intrinsic spatiotemporal locality in videos.

2022

84.9

Yang et al.

[100]

Fix the issue with videos' needed set length. A strategy that uses the attention date to repeatedly construct interaction between the current frame input and the prior hidden state.

107.7

0.11

2022

81.5

Xiong et al.

[101]

A multi-view transformer is composed of multiple individuals, each of which focuses on a certain depiction. Through a lateral link between individual encoders, information from several visual representations is successfully merged.

2022

89.1

Zha et al.

[102]

The components of the Shifted Chunk Transformer are a frame encoder and a clip encoder. The frame encoder uses the picture chunk and the shifting multi-head self-attention elements to capture intra-frame representation and inter-frame motion, respectively.

59.89

0.34

2021

98.7

84.6

83.0

Zhang et al. [103]

Using the proposed standard deviation, an approach that aggregates Spatial-temporal data with stacking attention and an attention-pooling strategy to reduce processing costs.

0.392

 0.39

2021

96.7

74.7

80.5

Hybrid- Model

Kalfaoglu et al.

[104]

Combining 3D convolution with late temporal modeling is a great way to improve the performance of 3D convolution designs.

94.74

0.07

2020

98.69

85.10

Bonan Li et al.

[105]

The issue of time-length videos was solved by implementing two attention modules: a short-term attention module and a long-term attention module, each of which provided a distinct temporal token attention.

2.17

2022

81.3

Table 2. RGB comparison of CNN, Transformer, and Hybrid model approaches for Action recognition on the UCF101 (UCF), HMDB51 (HDM), and Kinetics-400 (Kin) datasets in terms of accuracy and complexity [106, 107, 108]. [UCF] [106], [HMDB51 (HDM) [107], and [Kinetics-400 (Kin) [108] datasets (FLops)

7.2. Modality comparative literature

Visual data may originate from a range of modalities and may be utilised singly or in combination to describe a visual activity [109]. This review concentrates on the RGB data and the skeletal data. RGB data is commonly employed in real-world application scenarios since it is simple to acquire and provides a wealth of information [109].

Using the following RGB datasets, we evaluate the efficacy of the techniques under consideration: UCF101[106], which contains 27 hours of video data, over 13,000 clips, and 101 action classes, containing video data totaling over 13,000 minutes.

HMDB51[107] consists of 51 action categories and 7,000 manually annotated footage extracted from various sources, including digital movies and YouTube.

Kinetics-400 [42] includes 400 classes of human motion with at least 400 video clips for each class. Each clip lasts about 10 seconds. The accuracy metric is used to evaluate the models since all classes are similarly important.

In support of what has already been stated, Table 2 illustrates how performance and complexity vary from one model to the next by highlighting a variety of methodologies utilized for each RGB-based model.

In the Kinetics dataset, Xiong et al. [101] achieved the highest level of accuracy with 89.1% within the Google Research lab, exceeding the findings that have been considered to be state-of-the-art thus far. This in no way negates the fact that all of the models discussed produce findings that are fairly encouraging, independent of the datasets and metrics that are taken into consideration. Each model tries to fix a distinct issue by considering the existing issues that have been found.

For example, Bonan Li et al. [105], based on a CNN architecture, addressed the problem of time-length movies with two attention mechanisms. This confirms the hybrid model's viability as a method because it demonstrates that the problem can be solved using multiple attention mechanisms. Skeleton data is an additional acceptable modality for action recognition because it encodes the trajectories of human body joints. These trajectories reflect meaningful human movements, simplicity, and informative representation, which are the primary characteristics of skeleton data.

The NTU-RGB+D 60 [84] and the NTU-RGB+D 120 [85] datasets are the most well-known examples of 3D action recognition. NTU-RGB+D 6O is comprised of 56,880 video clips and 60 activities that were carried out by 40 different individuals. Each individual human skeleton is made up of 25 joints, each of which has a unique set of three-dimensional coordinates. The NTU-RGB+D 120 database is an expansion of the NTU-RGB+D 60 database, and it contains 114,480 skeletal sequences of 120 action classes performed by 106 unique subjects. Cross-Subject Evaluation (CS) and Cross-View Evaluation are the metrics that are utilized in order to evaluate the reported outcomes in relation to these two benchmarks (CV).

CS evaluates the model according to the subjects who participated in the data set, while CV analyses the model according to the camera view. The evaluation results are presented for each of them as the classification accuracy is expressed as a percentage. CNN [110] served as the inspiration for the creation of a graph convolutional network (GCN), which was proposed to capture the structural relations among the data.

Because skeleton data occurs naturally in graphs, numerous GCN techniques have been developed to represent skeleton data as graph structures consisting of edges and nodes. This is possible because skeleton data occurs naturally in the form of graphs. Since GCN uses convolutions, works that employ this technology have been placed in the same area as CNN.

Table 3 is quite similar to Table 2. Still, instead of comparing models using the same datasets and metrics as Table 2, it compares models using various datasets and metrics described earlier in this section. Recent research has shown the effectiveness of Transformer and self-attention in resolving the same challenge as skeleton-based action recognition. This is despite the fact that CNN and GCN have made significant strides in solving this problem.

Qin et al. [111] used the same graph model that was used in GCN; however, they added a technique that gave the Transformer model with both the joint and bone representations of the skeleton as a single input.

Model

Year

The idea of the model

Datasets (Accuracy)

NTU RGB+D

NTU RGB+D 120

CS

CV

CS

CV

CNN

Yan et al.

[112]

2018

The authors developed the first strategy for collecting spatial and temporal data from skeleton data by encoding the skeleton data with GCN.

85.7

92.7

Banerjee et al.

[113]

2020

The author developed a CNN-based classifier for each feature vector, combining Choquet fuzzy integral, Kull-back-Leibler, and Jensen-Shannon divergences to verify that the feature vectors are complementary.

84.2

89.7

74.8

76.9

Chen et al.

[114]

2021

The authors employed a GCN-based method to model dynamic channel-by-channel topologies employing a refining technique.

93.0

97.1

89.8

91.2

Chi et al.

[115]

2022

The authors developed a novel method that combines a learning objective and an encoding strategy. A learning objective based on the information bottleneck instructs the model to acquire informative, yet condensed, latent representations. To provide discriminative information, a multi-modal representation of the skeleton based on the relative positions of joints, an attention-based graph convolution that captures the context-dependent underlying topology of human activity, and complementing spatial information for joints.

92.1

96.1

88.7

88.9

Song et al.

[116]

2022

The authors developed a strategy based on a collection of GCN baselines to synchronously extend the width and depth of the model in order to extract discriminative features from all skeletal joints using a minimal number of trainable parameters.

92.1

96.1

88.7

88.9

Transformer

Shi et al.

[117]

2021

The authors designed a pure transformer model for peripheral platforms or real-time applications. Segmented linear self-attention module that records temporal correlations of dynamic joint motions and sparse self-attention module that performs sparse matrix multiplications to record spatial correlations between human skeletal joints.

83.4

84.2

78.3

78.5

Plizzari et al.

[82]

2021

The authors proposed a novel method to the modelling challenge posed by joint dependencies. A spatial self-attention (SSA) module is used to comprehend intra-frame interactions between various body parts, while a temporal self-attention (TSA) module is used to describe inter-frame correlations.

89.9

96.1

81.9

84.1

Helei et al.

[80]

2022

The authors propose two different modules. The first module records the interaction between multiple joints in consecutive frames, while the second module integrates the characteristics of various sub-action segments to record information about multiple joints between frames.

92.3

96.5

88.3

89.2

Hybrid models

Wang et al.

[111]

2021

The authors investigated employing a Transformer method to decrease the noise caused by operating joints. They suggested simultaneously encoding joint and body part interactions.

92.3

96.4

88.4

89.7

Qin et al.

[118]

2022

The authors proposed a strategy for concatenating the representation of joints and bones to the input layer using a single flow network in order to lower the computational cost.

90.5

96.1

85.7

86.8

Table 3. Skeleton modality comparison of CNN, Transformer, and Hybrid model techniques for action recognition on the datasets NTU-RGB+D 60[84] and NTU-RGB+D 120 [85] in terms of the accuracy according to the Cross-Subject Evaluation (CS) and Cross-View Evaluation (CV).

The conclusion has been changed accordingly.

New Conclusion

This work examines CNN and Transformer for Action Recognition individually, as well as the trade-off between accuracy and complexity. In addition, this paper evaluates the majority of pertinent research emphasizing the benefits of each of the aforementioned tactics and their corresponding outcomes.

The challenge of visual action recognition is fraught with obstacles and limits. Since the quality of research has improved over time, it is evident that solutions are on the horizon for addressing these issues, whether by employing CNN or Transformer approach. Transformer, which is fairly new to the field of computer vision, has been quite competitive with CNN, which is ten years more established up to this point.

As for the primary question, and in light of this study, it should be mentioned that although both algorithms (i.e., CNN and Transformers) work in their way and have their own shortcomings and benefits, it is still difficult to determine who will win this race. Nevertheless, the hybrid method that is more efficient and cost-effective. It combines CNN with transformers to provide a reliable model. After all, the old adage asserts that working together is the key to success!

This hybrid model is the most attractive formula because it enables us to take advantage of a model's strengths while simultaneously reducing the effects of that model's downsides. Additionally, it has been demonstrated that hybrid models are highly useful for bridging the gaps generated by the deficiencies of specific models.

Therefore, we believe that this hybrid model might win the race. Furthermore, we anticipate a greater emphasis on testing this hybrid approach in action recognitions in visual data

Comment 2:  The article describes in Section 6.3 that "despite the good results of Vision Transformer, the community still focuses on CNN structure. The article does not carry out an in-depth analysis on this point.

Author's response: We thank the reviewer for this comment. We have made the necessary changes to improve the results of our work.

Author's actions: We added new section, and modified the section 6.

Tables 2 and 3 in the new manuscript replace and update tables 1 and 2 in the previous document. In these new tables, we compared CNN, Transformer, and Hybrid-model approaches based on their modalities, whereas in the previous tables, we compared them based on their datasets. In the new table (Table 2), we list the most recent study completed on the RGB modality for each approach. For each study, we describe the concept underlying the model and its performance on the most well-known RGB datasets. This table also includes the execution time (in FLOPS) of each model, a feature added per your suggestion.

In the other new table (Table 3), the identical work was performed utilizing the skeletal modality as opposed to RGB. Nonetheless, the measurements and datasets employed in this instance differ from those in the preceding table. Instead of calculating precision and execution time, researchers in this field rely on the CS and CV measures.

The document has been updated to reflect the changes.

Reviewer 2 Report

The submitted manuscript is interesting and well written. However, the comparison part of the manuscript is rather superficial. In Subsection 6.1, the authors write about the performance of CNNs but not about the performance of Vision Transformers. In Subsection 6.2, the case is just the opposite. In Tables 1 and 2, citations are missing and these tables compare only four methods based on Vision Transformers. Moreover, an analysis about the execution time of CNN and Vision Transformers models is not presented. Conclusion is not drawn. The authors do not answer the question that they have formulated in the introduction section. Moreover, they repeat the question in the conclusion section. Unfortunately, the manuscript is not suitable for publication in its current form. Although it is a comparative study, it contains only a minimal amount of comparison. The reader is not convinced that the authors provide enough quantitative results which is usually the main goal of scientific publications in engineering fields.

Author Response

REVIEWER #2:

The submitted manuscript is interesting and well written. However, the comparison part of the manuscript is rather superficial. In Subsection 6.1, the authors write about the performance of CNNs but not about the performance of Vision Transformers. In Subsection 6.2, the case is just the opposite. In Tables 1 and 2, citations are missing and these tables compare only four methods based on Vision Transformers. Moreover, an analysis about the execution time of CNN and Vision Transformers models is not presented. Conclusion is not drawn. The authors do not answer the question that they have formulated in the introduction section. Moreover, they repeat the question in the conclusion section. Unfortunately, the manuscript is not suitable for publication in its current form. Although it is a comparative study, it contains only a minimal amount of comparison. The reader is not convinced that the authors provide enough quantitative results which is usually the main goal of scientific publications in engineering fields.

Dear reviewer, thank you for the time you took to study our work. After reading your constructive comments here are the modifications brought to the paper for each point.

Comment 1:  The comparison part of the manuscript is rather superficial.

Author's response: We thank the reviewer for this comment. We have made the necessary changes to improve the results of our work.

Author's actions: Indeed, the comparison was kind of superficial, we rectified this important point by adding two sections named "Pros and cons of CNN and Transformer”, and “Modality comparative literature”, respectively. These two sections 7.1 and 7.2 in the new manuscript replaced the old sections 6.1 and 6.2. Section 7.1 enhances the particularity of each method and section 7.2 presents a thorough comparison of recent works in the literature of each method.

To ensure a fair comparison, we use a multi-criteria analogy term in this section. We first present the advantages and disadvantages of the considered approaches, and then outline the state of the art of each method in terms of their modality, regarding the most widely used academic action recognition datasets.

7.1. Pros and cons of CNN and Transformer

One of the most significant advantages of CNN is the sparse connectivity that is provided by the Convolutional layer. This sparse connectivity saves memory because it only needs a small number of weights and connections [52], which makes it memory-effective. Because there are no weights allocated between two neurons of neighboring layers in the CNN, and the set of weights operates with one and all pixels of the input matrix [52], weight sharing is another essential feature of the convolutional layer. This feature helps reduce the time needed for training and the costs associated with training.

In their evaluation of previous work, Koot et al. [88] discovered that CNN performs better than Transformer when it comes to latency accuracy on lightweight datasets. CNN is also described to capture inductive bias, which is also known as prior knowledge, such as translation equivariance and localization while having pooling operation give partial scale invariance [89]. CNN, on the other hand, has a few weaknesses, including a slowness that is brought on by the max pooling operation, and, in contrast to Transformer, it does not take into consideration the several perspectives that can be gained by learning [90], which leads to a disregard for global knowledge.

Due to the fact that it offers solutions to CNN's numerous weaknesses, the Transformer has quickly become CNN's most formidable opponent. The capability of Transformer to prioritize relevant content while minimizing the repetition of content that is not important is one of the program's capabilities [89]. In addition, because less demand is placed on the processing power, the visual characteristics are dynamically re-weighted based on the context [90]. This ability to mimic long-distance interactions in space and time, an essential requirement for visual movies [91], is another reason the Transformer stands out in front of CNN.

The transformer effectively encodes temporal data, an additional crucial component for action recognition. Lastly, multi-head attention, which is the booster part of the performance of vanilla self-attention and the essential component of the visual transformer process, affords the opportunity to learn many representations from various perspectives [92].

In spite of this, Dosovitskiy et al. [11] found in their research that transformer has a significantly lower level of image-specific inductive bias compared to CNNs. In order to overcome inductive bias, the model needs to be trained on very large and sufficient datasets so that it can figure out these image-specific features on its own based on the training examples. Therefore, it is important for the self-attention mechanism to automatically recognize the temporal relationships between video frames by searching through a huge database of video sequences. This is done in order to fulfill the requirements of the self-attention mechanism. The end consequence is longer training timelines, a significant increase in the demands placed on computer resources, and enormous datasets that need to be processed [90].

In light of what has been said, researchers are progressively merging these two models into a single model in order to leverage the complementary strengths of the two models and offset the flaws of the two models. The findings of Zhao et al. [93] demonstrate that Transformer and CNN are mutually supportive of one another and could both be integrated into a single predictive model. They developed a hybrid model by employing multi-stage convolutional layers as the backbone of the model and exchanging a few particular layers for transformer layers. This offers the hybrid model with the global and local modelling capabilities of transformers and convolutions, respectively.

Dosovitskiy et al. [11] also acknowledged it. Experiments were undertaken to demonstrate that the transformer model excels after training on the CNN model. In light of CNN's ability to add location data to the Transformer model, it is important to note that the network is still in existence. Thus, a substantial amount of effort is required to add convolution to a typical transformer block. Hybrid models are approaches that mix CNN and Transformer.

Table 1 depicts the complementarity of the two models. A summary of the most pertinent works is offered in the subsequent section.

 Properties

CNN

Transformer

Hybrid model

Sparse connectivity

-

Weight sharing

-

Best at Latency accuracy on small
datasets

-

Inductive bias

-

Capture local information

-

Dynamic weight

-

Capture global information

-

Learn from different angles

-

Best at Spatio-temporal Model long-distance interactions

-

Table 1: CNN, Transformer, and hybrid model advantages,  “-“ indicates that the property is  invalid in the method.

Model + (Citation)

The idea of the model

Parameters

Flops

Year

Datasets (Accuracy)

UCF

HDM

Kin

CNN

Omi et al.

[94]

Present a 3D CNN multi-domain-learning using adapters between layers. The results showed with ResNet backbone.

183.34

2022

63.33

93.29

67.80

TEINet

[95]

A technique for temporal modeling that enhances motion-related properties and adjusts the temporal contextual information channel-wise (backbone ResNet-50).

0.06

2020

96.7

72.1

76.2

Xinyu Li

[96]

Introduced a 3D CNN network that learns video clip-level temporal features from different spatial and temporal scales.

103

0.12

2020

97.9

75.2

SlowFast Networks

[48]

A single-stream design that operates at two separate frame rates. SlowPath captures spatial semantics, but FastPath combines temporal semantics via the side connection. We displayed the outcomes using a 3D Resnet backbone.

32.88

0.36

2019

75.6

Du Tran et al.

[97]

Suggested factorizing 3D convolutions by separating channel interactions and spatiotemporal interactions in order to obtain greater precision at a lower computing cost.

32.8

1.08

2019

82.5

Transformer

Jue Wang1 et al. [98]

Dynamically predicts a subset of video patches to attend for each query location based on motion information.

73.9M

1.28

2022

79.0

Arnab et al.

[81]

Presented multiple models which factorize different components of the spatial-temporal transformer encoder. A solution to regularize the transformer model during training small datasets.

4.77

2021

84.9

Liu et al.

[99]

A pure transformer backbone model addressed the inductive bias of locality by utilizing the advantage of the intrinsic spatiotemporal locality in videos.

2022

84.9

Yang et al.

[100]

Fix the issue with videos' needed set length. A strategy that uses the attention date to repeatedly construct interaction between the current frame input and the prior hidden state.

107.7

0.11

2022

81.5

Xiong et al.

[101]

A multi-view transformer is composed of multiple individuals, each of which focuses on a certain depiction. Through a lateral link between individual encoders, information from several visual representations is successfully merged.

2022

89.1

Zha et al.

[102]

The components of the Shifted Chunk Transformer are a frame encoder and a clip encoder. The frame encoder uses the picture chunk and the shifting multi-head self-attention elements to capture intra-frame representation and inter-frame motion, respectively.

59.89

0.34

2021

98.7

84.6

83.0

Zhang et al. [103]

Using the proposed standard deviation, an approach that aggregates Spatial-temporal data with stacking attention and an attention-pooling strategy to reduce processing costs.

0.392

 0.39

2021

96.7

74.7

80.5

Hybrid- Model

Kalfaoglu et al.

[104]

Combining 3D convolution with late temporal modeling is a great way to improve the performance of 3D convolution designs.

94.74

0.07

2020

98.69

85.10

Bonan Li et al.

[105]

The issue of time-length videos was solved by implementing two attention modules: a short-term attention module and a long-term attention module, each of which provided a distinct temporal token attention.

2.17

2022

81.3

Table 2. RGB comparison of CNN, Transformer, and Hybrid model approaches for Action recognition on the UCF101 (UCF), HMDB51 (HDM), and Kinetics-400 (Kin) datasets in terms of accuracy and complexity [106, 107, 108]. [UCF] [106], [HMDB51 (HDM) [107], and [Kinetics-400 (Kin) [108] datasets (FLops)

7.2. Modality comparative literature

Visual data may originate from a range of modalities and may be utilised singly or in combination to describe a visual activity [109]. This review concentrates on the RGB data and the skeletal data. RGB data is commonly employed in real-world application scenarios since it is simple to acquire and provides a wealth of information [109].

Using the following RGB datasets, we evaluate the efficacy of the techniques under consideration: UCF101[106], which contains 27 hours of video data, over 13,000 clips, and 101 action classes, containing video data totaling over 13,000 minutes.

HMDB51[107] consists of 51 action categories and 7,000 manually annotated footage extracted from various sources, including digital movies and YouTube.

Kinetics-400 [42] includes 400 classes of human motion with at least 400 video clips for each class. Each clip lasts about 10 seconds. The accuracy metric is used to evaluate the models since all classes are similarly important.

In support of what has already been stated, Table 2 illustrates how performance and complexity vary from one model to the next by highlighting a variety of methodologies utilized for each RGB-based model.

In the Kinetics dataset, Xiong et al. [101] achieved the highest level of accuracy with 89.1% within the Google Research lab, exceeding the findings that have been considered to be state-of-the-art thus far. This in no way negates the fact that all of the models discussed produce findings that are fairly encouraging, independent of the datasets and metrics that are taken into consideration. Each model tries to fix a distinct issue by considering the existing issues that have been found.

For example, Bonan Li et al. [105], based on a CNN architecture, addressed the problem of time-length movies with two attention mechanisms. This confirms the hybrid model's viability as a method because it demonstrates that the problem can be solved using multiple attention mechanisms. Skeleton data is an additional acceptable modality for action recognition because it encodes the trajectories of human body joints. These trajectories reflect meaningful human movements, simplicity, and informative representation, which are the primary characteristics of skeleton data.

The NTU-RGB+D 60 [84] and the NTU-RGB+D 120 [85] datasets are the most well-known examples of 3D action recognition. NTU-RGB+D 6O is comprised of 56,880 video clips and 60 activities that were carried out by 40 different individuals. Each individual human skeleton is made up of 25 joints, each of which has a unique set of three-dimensional coordinates. The NTU-RGB+D 120 database is an expansion of the NTU-RGB+D 60 database, and it contains 114,480 skeletal sequences of 120 action classes performed by 106 unique subjects. Cross-Subject Evaluation (CS) and Cross-View Evaluation are the metrics that are utilized in order to evaluate the reported outcomes in relation to these two benchmarks (CV).

CS evaluates the model according to the subjects who participated in the data set, while CV analyses the model according to the camera view. The evaluation results are presented for each of them as the classification accuracy is expressed as a percentage. CNN [110] served as the inspiration for the creation of a graph convolutional network (GCN), which was proposed to capture the structural relations among the data.

Because skeleton data occurs naturally in graphs, numerous GCN techniques have been developed to represent skeleton data as graph structures consisting of edges and nodes. This is possible because skeleton data occurs naturally in the form of graphs. Since GCN uses convolutions, works that employ this technology have been placed in the same area as CNN.

Table 3 is quite similar to Table 2. Still, instead of comparing models using the same datasets and metrics as Table 2, it compares models using various datasets and metrics described earlier in this section. Recent research has shown the effectiveness of Transformer and self-attention in resolving the same challenge as skeleton-based action recognition. This is despite the fact that CNN and GCN have made significant strides in solving this problem.

Qin et al. [111] used the same graph model that was used in GCN; however, they added a technique that gave the Transformer model with both the joint and bone representations of the skeleton as a single input.

Model

Year

The idea of the model

Datasets (Accuracy)

NTU RGB+D

NTU RGB+D 120

CS

CV

CS

CV

CNN

Yan et al.

[112]

2018

The authors developed the first strategy for collecting spatial and temporal data from skeleton data by encoding the skeleton data with GCN.

85.7

92.7

Banerjee et al.

[113]

2020

The author developed a CNN-based classifier for each feature vector, combining Choquet fuzzy integral, Kull-back-Leibler, and Jensen-Shannon divergences to verify that the feature vectors are complementary.

84.2

89.7

74.8

76.9

Chen et al.

[114]

2021

The authors employed a GCN-based method to model dynamic channel-by-channel topologies employing a refining technique.

93.0

97.1

89.8

91.2

Chi et al.

[115]

2022

The authors developed a novel method that combines a learning objective and an encoding strategy. A learning objective based on the information bottleneck instructs the model to acquire informative, yet condensed, latent representations. To provide discriminative information, a multi-modal representation of the skeleton based on the relative positions of joints, an attention-based graph convolution that captures the context-dependent underlying topology of human activity, and complementing spatial information for joints.

92.1

96.1

88.7

88.9

Song et al.

[116]

2022

The authors developed a strategy based on a collection of GCN baselines to synchronously extend the width and depth of the model in order to extract discriminative features from all skeletal joints using a minimal number of trainable parameters.

92.1

96.1

88.7

88.9

Transformer

Shi et al.

[117]

2021

The authors designed a pure transformer model for peripheral platforms or real-time applications. Segmented linear self-attention module that records temporal correlations of dynamic joint motions and sparse self-attention module that performs sparse matrix multiplications to record spatial correlations between human skeletal joints.

83.4

84.2

78.3

78.5

Plizzari et al.

[82]

2021

The authors proposed a novel method to the modelling challenge posed by joint dependencies. A spatial self-attention (SSA) module is used to comprehend intra-frame interactions between various body parts, while a temporal self-attention (TSA) module is used to describe inter-frame correlations.

89.9

96.1

81.9

84.1

Helei et al.

[80]

2022

The authors propose two different modules. The first module records the interaction between multiple joints in consecutive frames, while the second module integrates the characteristics of various sub-action segments to record information about multiple joints between frames.

92.3

96.5

88.3

89.2

Hybrid models

Wang et al.

[111]

2021

The authors investigated employing a Transformer method to decrease the noise caused by operating joints. They suggested simultaneously encoding joint and body part interactions.

92.3

96.4

88.4

89.7

Qin et al.

[118]

2022

The authors proposed a strategy for concatenating the representation of joints and bones to the input layer using a single flow network in order to lower the computational cost.

90.5

96.1

85.7

86.8

Table 3. Skeleton modality comparison of CNN, Transformer, and Hybrid model techniques for action recognition on the datasets NTU-RGB+D 60[84] and NTU-RGB+D 120 [85] in terms of the accuracy according to the Cross-Subject Evaluation (CS) and Cross-View Evaluation (CV).

Comment 2:  In Subsection 6.1, the authors write about the performance of CNNs but not about the performance of Vision Transformers. In Subsection 6.2, the case is just the opposite. 

Author's response: We thank the reviewer for this comment. We have made the necessary changes to improve the results of our work.

Author's actions:

In the same way that we improved the comparison part, in sections 7.1 and 7.2 (which are replacing the old sections 6.1 and 6.2), we operated differently; first, we carefully considered the benefits and drawbacks of each approach (CNN, transformers), and then we compiled these details into a table (Table 1), contrasting them and highlighting the function of hybrid models in handling and resolving the issues that each approach faced. In other words, we improved the comparison part by operating differently.

Our review of the bibliography has been improved thanks to the addition of fresh references (118 references). Due to the novelty of the subject matter, we have concentrated only on more recent publications (mostly from 2021 and 2022).

Comment 3:  In Tables 1 and 2, citations are missing and these tables compare only four methods based on Vision Transformers. Moreover, an analysis about the execution time of CNN and Vision Transformers models is not presented.

Author's response: We thank the reviewer for this comment. We have made the necessary changes to improve the results of our work.

Author's actions:

Tables 2 and 3 in the new manuscript replace and update tables 1 and 2 in the previous document. In these new tables, we compared CNN, Transformer, and Hybrid-model approaches based on their modalities, whereas in the previous tables, we compared them based on their datasets. In the new table (Table 2), we list the most recent study completed on the RGB modality for each approach. For each study, we describe the concept underlying the model and its performance on the most well-known RGB datasets. This table also includes the execution time (in FLOPS) of each model, a feature added per your suggestion.

In the other new table (Table 3), the identical work was performed utilizing the skeletal modality as opposed to RGB. Nonetheless, the measurements and datasets employed in this instance differ from those in the preceding table. Instead of calculating precision and execution time, researchers in this field rely on the CS and CV measures.

The document has been revised.

Comment 4: Conclusion is not drawn. The authors do not answer the question that they have formulated in the introduction section. Moreover, they repeat the question in the conclusion section. Unfortunately, the manuscript is not suitable for publication in its current form.

Author's response: We thank the reviewer for this comment. We have made the necessary changes to improve the results of our work.

Author's actions: The revised version of the conclusion now makes it more specific. The basic question posed in the introduction is now addressed throughout the comparison section and reiterated in the conclusion.

The conclusion has been changed accordingly.

New Conclusion:

This work examines CNN and Transformer for Action Recognition individually, as well as the trade-off between accuracy and complexity. In addition, this paper evaluates the majority of pertinent research emphasizing the benefits of each of the aforementioned tactics and their corresponding outcomes.

The challenge of visual action recognition is fraught with obstacles and limits. Since the quality of research has improved over time, it is evident that solutions are on the horizon for addressing these issues, whether by employing CNN or Transformer approach. Transformer, which is fairly new to the field of computer vision, has been quite competitive with CNN, which is ten years more established up to this point.

As for the primary question, and in light of this study, it should be mentioned that although both algorithms (i.e., CNN and Transformers) work in their way and have their own shortcomings and benefits, it is still difficult to determine who will win this race. Nevertheless, the hybrid method that is more efficient and cost-effective. It combines CNN with transformers to provide a reliable model. After all, the old adage asserts that working together is the key to success!

This hybrid model is the most attractive formula because it enables us to take advantage of a model's strengths while simultaneously reducing the effects of that model's downsides. Additionally, it has been demonstrated that hybrid models are highly useful for bridging the gaps generated by the deficiencies of specific models.

Therefore, we believe that this hybrid model might win the race. Furthermore, we anticipate a greater emphasis on testing this hybrid approach in action recognitions in visual data

Comment 5: Although it is a comparative study, it contains only a minimal amount of comparison. The reader is not convinced that the authors provide enough quantitative results which is usually the main goal of scientific publications in engineering fields.

Author's response: We thank the reviewer for this suggestion. We have made the necessary changes to improve the results of our work.

Author’s actions: We were able to solve the problem of quantitative and qualitative lack of comparison by implementing your suggestion and enhancing the entire paper. These enhancements are found in sections 7 and 8.

Reviewer 3 Report

Summary

This paper presents a survey of CNN and Transformers in the application of action recognition (AR) in visual data. Overall, many existing models are included with brief summaries, and the comparisons between two lines of research are explored. On the other hand, the takeaways of the comparisons are not clear, and the conclusion of this manuscript is unclear, leaving the question in the survey title, "who will win the race for action recognitions in visual data" not fully answered.

While the flow of this paper is clear, there are several issues with it before being considered acceptance.

Major comments:

1. The real meat of this survey is the comparison between two lines of research: CNN and Transformers. Unfortunately, only the results of GCN-based methods are illustrated. How about the performance of models such as more "classic" CNN architectures, Resnet, etc.?

2. The statements in section 6 are unclear and lack takeaways. It is hard to understand what to get from this comparison.

3. The primary question indicated in the title, "who will win the race for action recognitions in visual data" is not fully answered in this manuscript, it is recommended to provide more insights such as under what conditions, or for which datasets, CNN/Transformers tend to outperform.

Author Response

REVIEWER #3:

This paper presents a survey of CNN and Transformers in the application of action recognition (AR) in visual data. Overall, many existing models are included with brief summaries, and the comparisons between two lines of research are explored. On the other hand, the takeaways of the comparisons are not clear, and the conclusion of this manuscript is unclear, leaving the question in the survey title, "who will win the race for action recognitions in visual data" not fully answered.

While the flow of this paper is clear, there are several issues with it before being considered acceptance.

Dear reviewer, thank you for the time you took to study our work. After reading your constructive comments here are the modifications brought to the paper for each point.

 Major comments:

Comment 1: The real meat of this survey is the comparison between two lines of research: CNN and Transformers. Unfortunately, only the results of GCN-based methods are illustrated. How about the performance of models such as more "classic" CNN architectures, Resnet, etc.?

Author's response: We thank the reviewer for this comment. We have made the necessary changes to improve the results of our work.

Author's actions: We added the following section

In 2015, Tran et al. presented a study [52] to find the most appropriate temporal length for 3D CNN and developed a VGG-style 3D CNN architecture. The same authors performed a search for 3D CNN in a deep residual learning framework and employed a ResNet18 style 3D CNN architecture named Res3D; this surpasses C3D by a considerable margin in terms of accuracy and recognition time.

And enhanced the following figure

Figure 2: The evolution of CNN over time (Inception [21], ZefNet[22], ResNet [23], MobileNet[24], and HRNet [25] )

We added the table 2 with very detailed  comparaison

Table 2. RGB comparison of CNN, Transformer, and Hybrid model approaches for Action recognition on the UCF101 (UCF), HMDB51 (HDM), and Kinetics-400 (Kin) datasets in terms of accuracy and complexity [106, 107, 108]. [UCF] [106], [HMDB51 (HDM) [107], and [Kinetics-400 (Kin) [108] datasets (FLops)

  1. The statements in section 6 are unclear and lack takeaways. It is hard to understand what to get from this comparison.

Author's response: We thank the reviewer for this comment. We have made the necessary changes to improve the results of our work.

Author's actions: Indeed, the comparison was kind of superficial, we rectified this important point by adding two sections named "Pros and cons of CNN and Transformer”, and “Modality comparative literature”, respectively. These two sections 7.1 and 7.2 in the new manuscript replaced the old sections 6.1 and 6.2. Section 7.1 enhances the particularity of each method and section 7.2 presents a thorough comparison of recent works in the literature of each method.

  1. The primary question indicated in the title, "who will win the race for action recognitions in visual data" is not fully answered in this manuscript, it is recommended to provide more insights such as under what conditions, or for which datasets, CNN/Transformers tend to outperform.

Author's response: We thank the reviewer for this comment. We have made the necessary changes to improve the results of our work.

Author's actions:

WE updated the conclusion, with answering the primary question of the paper.

New Conclusion

This work examines CNN and Transformer for Action Recognition individually, as well as the trade-off between accuracy and complexity. In addition, this paper evaluates the majority of pertinent research emphasizing the benefits of each of the aforementioned tactics and their corresponding outcomes.

The challenge of visual action recognition is fraught with obstacles and limits. Since the quality of research has improved over time, it is evident that solutions are on the horizon for addressing these issues, whether by employing CNN or Transformer approach. Transformer, which is fairly new to the field of computer vision, has been quite competitive with CNN, which is ten years more established up to this point.

As for the primary question, and in light of this study, it should be mentioned that although both algorithms (i.e., CNN and Transformers) work in their way and have their own shortcomings and benefits, it is still difficult to determine who will win this race. Nevertheless, the hybrid method that is more efficient and cost-effective. It combines CNN with transformers to provide a reliable model. After all, the old adage asserts that working together is the key to success!

This hybrid model is the most attractive formula because it enables us to take advantage of a model's strengths while simultaneously reducing the effects of that model's downsides. Additionally, it has been demonstrated that hybrid models are highly useful for bridging the gaps generated by the deficiencies of specific models.

Therefore, we believe that this hybrid model might win the race. Furthermore, we anticipate a greater emphasis on testing this hybrid approach in action recognitions in visual data.

Furthermore, we added more details in the revised version of the paper.

Please see the added table and discussion.

7.1. Pros and cons of CNN and Transformer

One of the most significant advantages of CNN is the sparse connectivity that is provided by the Convolutional layer. This sparse connectivity saves memory because it only needs a small number of weights and connections [52], which makes it memory-effective. Because there are no weights allocated between two neurons of neighboring layers in the CNN, and the set of weights operates with one and all pixels of the input matrix [52], weight sharing is another essential feature of the convolutional layer. This feature helps reduce the time needed for training and the costs associated with training.

In their evaluation of previous work, Koot et al. [88] discovered that CNN performs better than Transformer when it comes to latency accuracy on lightweight datasets. CNN is also described to capture inductive bias, which is also known as prior knowledge, such as translation equivariance and localization while having pooling operation give partial scale invariance [89].

CNN, on the other hand, has a few weaknesses, including a slowness that is brought on by the max pooling operation, and, in contrast to Transformer, it does not take into consideration the several perspectives that can be gained by learning [90], which leads to a disregard for global knowledge.

Due to the fact that it offers solutions to CNN's numerous weaknesses, the Transformer has quickly become CNN's most formidable opponent. The capability of Transformer to prioritize relevant content while minimizing the repetition of content that is not important is one of the program's capabilities [89]. In addition, because less demand is placed on the processing power, the visual characteristics are dynamically re-weighted based on the context [90]. This ability to mimic long-distance interactions in space and time, an essential requirement for visual movies [91], is another reason the Transformer stands out in front of CNN.

The transformer effectively encodes temporal data, an additional crucial component for action recognition. Lastly, multi-head attention, which is the booster part of the performance of vanilla self-attention and the essential component of the visual transformer process, affords the opportunity to learn many representations from various perspectives [92].

In spite of this, Dosovitskiy et al. [11] found in their research that transformer has a significantly lower level of image-specific inductive bias compared to CNNs. In order to overcome inductive bias, the model needs to be trained on very large and sufficient datasets so that it can figure out these image-specific features on its own based on the training examples. Therefore, it is important for the self-attention mechanism to automatically recognize the temporal relationships between video frames by searching through a huge database of video sequences. This is done in order to fulfill the requirements of the self-attention mechanism. The end consequence is longer training timelines, a significant increase in the demands placed on computer resources, and enormous datasets that need to be processed [90].

In light of what has been said, researchers are progressively merging these two models into a single model in order to leverage the complementary strengths of the two models and offset the flaws of the two models.

The findings of Zhao et al. [93] demonstrate that Transformer and CNN are mutually supportive of one another and could both be integrated into a single predictive model. They developed a hybrid model by employing multi-stage convolutional layers as the backbone of the model and exchanging a few particular layers for transformer layers. This offers the hybrid model with the global and local modelling capabilities of transformers and convolutions, respectively.

Dosovitskiy et al. [11] also acknowledged it. Experiments were undertaken to demonstrate that the transformer model excels after training on the CNN model. In light of CNN's ability to add location data to the Transformer model, it is important to note that the network is still in existence. Thus, a substantial amount of effort is required to add convolution to a typical transformer block. Hybrid models are approaches that mix CNN and Transformer.

Table 1 depicts the complementarity of the two models. A summary of the most pertinent works is offered in the subsequent section.

 Properties

CNN

Transformer

Hybrid model

Sparse connectivity

-

Weight sharing

-

Best at Latency accuracy on small
datasets

-

Inductive bias

-

Capture local information

-

Dynamic weight

-

Capture global information

-

Learn from different angles

-

Best at Spatio-temporal Model long-distance interactions

-

Table 1: CNN, Transformer, and hybrid model advantages,  “-“ indicates that the property is  invalid in the method.

Model + (Citation)

The idea of the model

Parameters

Flops

Year

Datasets (Accuracy)

UCF

HDM

Kin

CNN

Omi et al.

[94]

Present a 3D CNN multi-domain-learning using adapters between layers. The results showed with ResNet backbone.

183.34

2022

63.33

93.29

67.80

TEINet

[95]

A technique for temporal modeling that enhances motion-related properties and adjusts the temporal contextual information channel-wise (backbone ResNet-50).

0.06

2020

96.7

72.1

76.2

Xinyu Li

[96]

Introduced a 3D CNN network that learns video clip-level temporal features from different spatial and temporal scales.

103

0.12

2020

97.9

75.2

SlowFast Networks

[48]

A single-stream design that operates at two separate frame rates. SlowPath captures spatial semantics, but FastPath combines temporal semantics via the side connection. We displayed the outcomes using a 3D Resnet backbone.

32.88

0.36

2019

75.6

Du Tran et al.

[97]

Suggested factorizing 3D convolutions by separating channel interactions and spatiotemporal interactions in order to obtain greater precision at a lower computing cost.

32.8

1.08

2019

82.5

Transformer

Jue Wang1 et al. [98]

Dynamically predicts a subset of video patches to attend for each query location based on motion information.

73.9M

1.28

2022

79.0

Arnab et al.

[81]

Presented multiple models which factorize different components of the spatial-temporal transformer encoder. A solution to regularize the transformer model during training small datasets.

4.77

2021

84.9

Liu et al.

[99]

A pure transformer backbone model addressed the inductive bias of locality by utilizing the advantage of the intrinsic spatiotemporal locality in videos.

2022

84.9

Yang et al.

[100]

Fix the issue with videos' needed set length. A strategy that uses the attention date to repeatedly construct interaction between the current frame input and the prior hidden state.

107.7

0.11

2022

81.5

Xiong et al.

[101]

A multi-view transformer is composed of multiple individuals, each of which focuses on a certain depiction. Through a lateral link between individual encoders, information from several visual representations is successfully merged.

2022

89.1

Zha et al.

[102]

The components of the Shifted Chunk Transformer are a frame encoder and a clip encoder. The frame encoder uses the picture chunk and the shifting multi-head self-attention elements to capture intra-frame representation and inter-frame motion, respectively.

59.89

0.34

2021

98.7

84.6

83.0

Zhang et al. [103]

Using the proposed standard deviation, an approach that aggregates Spatial-temporal data with stacking attention and an attention-pooling strategy to reduce processing costs.

0.392

 0.39

2021

96.7

74.7

80.5

Hybrid- Model

Kalfaoglu et al.

[104]

Combining 3D convolution with late temporal modeling is a great way to improve the performance of 3D convolution designs.

94.74

0.07

2020

98.69

85.10

Bonan Li et al.

[105]

The issue of time-length videos was solved by implementing two attention modules: a short-term attention module and a long-term attention module, each of which provided a distinct temporal token attention.

2.17

2022

81.3

Table 2. RGB comparison of CNN, Transformer, and Hybrid model approaches for Action recognition on the UCF101 (UCF), HMDB51 (HDM), and Kinetics-400 (Kin) datasets in terms of accuracy and complexity [106, 107, 108]. [UCF] [106], [HMDB51 (HDM) [107], and [Kinetics-400 (Kin) [108] datasets (FLops)

7.2. Modality comparative literature

Visual data may originate from a range of modalities and may be utilised singly or in combination to describe a visual activity [109]. This review concentrates on the RGB data and the skeletal data. RGB data is commonly employed in real-world application scenarios since it is simple to acquire and provides a wealth of information [109].

Using the following RGB datasets, we evaluate the efficacy of the techniques under consideration: UCF101[106], which contains 27 hours of video data, over 13,000 clips, and 101 action classes, containing video data totaling over 13,000 minutes.

HMDB51[107] consists of 51 action categories and 7,000 manually annotated footage extracted from various sources, including digital movies and YouTube.

Kinetics-400 [42] includes 400 classes of human motion with at least 400 video clips for each class. Each clip lasts about 10 seconds. The accuracy metric is used to evaluate the models since all classes are similarly important.

In support of what has already been stated, Table 2 illustrates how performance and complexity vary from one model to the next by highlighting a variety of methodologies utilized for each RGB-based model.

In the Kinetics dataset, Xiong et al. [101] achieved the highest level of accuracy with 89.1% within the Google Research lab, exceeding the findings that have been considered to be state-of-the-art thus far. This in no way negates the fact that all of the models discussed produce findings that are fairly encouraging, independent of the datasets and metrics that are taken into consideration. Each model tries to fix a distinct issue by considering the existing issues that have been found.

For example, Bonan Li et al. [105], based on a CNN architecture, addressed the problem of time-length movies with two attention mechanisms. This confirms the hybrid model's viability as a method because it demonstrates that the problem can be solved using multiple attention mechanisms. Skeleton data is an additional acceptable modality for action recognition because it encodes the trajectories of human body joints. These trajectories reflect meaningful human movements, simplicity, and informative representation, which are the primary characteristics of skeleton data.

The NTU-RGB+D 60 [84] and the NTU-RGB+D 120 [85] datasets are the most well-known examples of 3D action recognition. NTU-RGB+D 6O is comprised of 56,880 video clips and 60 activities that were carried out by 40 different individuals. Each individual human skeleton is made up of 25 joints, each of which has a unique set of three-dimensional coordinates. The NTU-RGB+D 120 database is an expansion of the NTU-RGB+D 60 database, and it contains 114,480 skeletal sequences of 120 action classes performed by 106 unique subjects. Cross-Subject Evaluation (CS) and Cross-View Evaluation are the metrics that are utilized in order to evaluate the reported outcomes in relation to these two benchmarks (CV).

CS evaluates the model according to the subjects who participated in the data set, while CV analyses the model according to the camera view. The evaluation results are presented for each of them as the classification accuracy is expressed as a percentage. CNN [110] served as the inspiration for the creation of a graph convolutional network (GCN), which was proposed to capture the structural relations among the data.

Because skeleton data occurs naturally in graphs, numerous GCN techniques have been developed to represent skeleton data as graph structures consisting of edges and nodes. This is possible because skeleton data occurs naturally in the form of graphs. Since GCN uses convolutions, works that employ this technology have been placed in the same area as CNN.

Table 3 is quite similar to Table 2. Still, instead of comparing models using the same datasets and metrics as Table 2, it compares models using various datasets and metrics described earlier in this section. Recent research has shown the effectiveness of Transformer and self-attention in resolving the same challenge as skeleton-based action recognition. This is despite the fact that CNN and GCN have made significant strides in solving this problem.

Qin et al. [111] used the same graph model that was used in GCN; however, they added a technique that gave the Transformer model with both the joint and bone representations of the skeleton as a single input.

Model

Year

The idea of the model

Datasets (Accuracy)

NTU RGB+D

NTU RGB+D 120

CS

CV

CS

CV

CNN

Yan et al.

[112]

2018

The authors developed the first strategy for collecting spatial and temporal data from skeleton data by encoding the skeleton data with GCN.

85.7

92.7

Banerjee et al.

[113]

2020

The author developed a CNN-based classifier for each feature vector, combining Choquet fuzzy integral, Kull-back-Leibler, and Jensen-Shannon divergences to verify that the feature vectors are complementary.

84.2

89.7

74.8

76.9

Chen et al.

[114]

2021

The authors employed a GCN-based method to model dynamic channel-by-channel topologies employing a refining technique.

93.0

97.1

89.8

91.2

Chi et al.

[115]

2022

The authors developed a novel method that combines a learning objective and an encoding strategy. A learning objective based on the information bottleneck instructs the model to acquire informative, yet condensed, latent representations. To provide discriminative information, a multi-modal representation of the skeleton based on the relative positions of joints, an attention-based graph convolution that captures the context-dependent underlying topology of human activity, and complementing spatial information for joints.

92.1

96.1

88.7

88.9

Song et al.

[116]

2022

The authors developed a strategy based on a collection of GCN baselines to synchronously extend the width and depth of the model in order to extract discriminative features from all skeletal joints using a minimal number of trainable parameters.

92.1

96.1

88.7

88.9

Transformer

Shi et al.

[117]

2021

The authors designed a pure transformer model for peripheral platforms or real-time applications. Segmented linear self-attention module that records temporal correlations of dynamic joint motions and sparse self-attention module that performs sparse matrix multiplications to record spatial correlations between human skeletal joints.

83.4

84.2

78.3

78.5

Plizzari et al.

[82]

2021

The authors proposed a novel method to the modelling challenge posed by joint dependencies. A spatial self-attention (SSA) module is used to comprehend intra-frame interactions between various body parts, while a temporal self-attention (TSA) module is used to describe inter-frame correlations.

89.9

96.1

81.9

84.1

Helei et al.

[80]

2022

The authors propose two different modules. The first module records the interaction between multiple joints in consecutive frames, while the second module integrates the characteristics of various sub-action segments to record information about multiple joints between frames.

92.3

96.5

88.3

89.2

Hybrid models

Wang et al.

[111]

2021

The authors investigated employing a Transformer method to decrease the noise caused by operating joints. They suggested simultaneously encoding joint and body part interactions.

92.3

96.4

88.4

89.7

Qin et al.

[118]

2022

The authors proposed a strategy for concatenating the representation of joints and bones to the input layer using a single flow network in order to lower the computational cost.

90.5

96.1

85.7

86.8

Table 3. Skeleton modality comparison of CNN, Transformer, and Hybrid model techniques for action recognition on the datasets NTU-RGB+D 60[84] and NTU-RGB+D 120 [85] in terms of the accuracy according to the Cross-Subject Evaluation (CS) and Cross-View Evaluation (CV).

Reviewer 4 Report

The paper presents an interesting idea but should be modified in some of its parts:

- Section 2 should be integrated with more recent works regarding the state of the art;

- Figure 6 is important. It should be better described in the caption and contextualized in the text;

- In the experimental part, measures more sensitive to data imbalance should be adopted, such as Matthew's correlation coefficient or balanced accuracy:

- The datasets used should be described perhaps in tabular form;

- With reference to figure 6, what happens if we try to reduce the dimensionality of space? In the specific context, the following paper should be cited:

Manipur, I., Manzo, M., Granata, I., Giordano, M., Maddalena, L., & Guarracino, M. R. (2021). Netpro2vec: a graph embedding framework for biomedical applications. IEEE/ACM Transactions on Computational Biology and Bioinformatics19(2), 729-740.

Author Response

REVIEWER #4:

Dear reviewer, thank you for the time you took to study our work. After reading your constructive comments here are the modifications brought to the paper for each point.

Comment 1:   Section 2 should be integrated with more recent works regarding the state of the art.

Author's response: We thank the reviewer for this comment. We have made the necessary changes to improve the results of our work.

Author's actions:  In response to your excellent suggestion, we included an entire section (section 3) to the updated manuscript. This section comprises recent and pertinent CNN-related research, as well as a graphic (Figure 4) demonstrating the topic's prominence over the years based on the number of studies conducted on it. In the same manner, to have a more systematic literature review, we included a comparable figure (Figure 6) regarding Transformer to Section 5.

  1. CNN-related Work

Figure 4: The distribution of CNN techniques in WOS on an annual basis. Since 2015, the number of studies on CNN applications in WOS has increased significantly.

In the past seven years, numerous studies and applications based on CNN have been developed in various domains, including healthcare [26] and autonomous vehicles [27].

Based on Web of Science databases, Figure 4 depicts the number of publications per year that mention CNN. Object detection is one of these applications, with the goal of identifying the object with a bounding box and determining its classification.

One-stage processes, such as YOLO [28], [29], SSD [30], and CornerNet [31]-[32], can be distinguished from two-stage techniques, such as R-CNN[33], Fast R-CNN[34], and Faster R-CNN[35], which made a breakthrough in object detection.

In the two-stage object detection process, region proposals are picked beforehand, and then CNN classifies the items. In a single step, the model simultaneously returns the category probability and position coordinates of the objects.

CNN[36], a biometric identification approach based on characteristics of the human face, is currently employed in the real world for face detection, a significant application.

Deepface [37] and DeepID [38] performed better than humans in unconstrained circumstances for the first time using the LFW dataset [39].

Detection, alignment, extraction, and classification constitute the DeepFace procedure. After detecting the face as the input to the CNN, Taigman et al. [37] trained the model using the Siamese network, achieving state-of-the-art performance. While DeepID directly enters two face pictures into CNN to extract feature vectors for classification, CNN is used by DeepID to extract feature vectors. Recent advancements in face recognition have primarily concentrated on the loss function.

FaceNet [38], which was proposed by Google in 2015, employs a 22-layer CNN to train a model with 200 million photos, including eight million humans. FaceNet substitutes softmax with triplet loss to discover more effective embeddings.

  1. Vision Transformer

In that respect, a 2D CNN model for image-level feature extraction and an additional model for temporal information capture. For example, the TRN [55] method relies on many features to structure images' relationships. The TSM[45] approach shifts some channels across the temporal dimension, allowing the exchange of information between neighboring images, or the TAM [40] method, which relies on 1 × 1 convolution in depth to capture temporal dependencies between images efficiently.

Various methods of temporal aggregation of feature descriptors have also been proposed. There are also more complex approaches that have also been studied on how to capture the long term. These models have achieved SOTA performance on multiple large-scale benchmarks, for instance, Kinetics [42], Something-Something [56], Sports1M [54], and Youtube-8M [57]. However, there is no winner between these approaches; 3D models perform more sufficiently than 2D standards on the Kinetics dataset, while 2D methods perform better on Something-Something.

  1. Vision Transformer

Figure 6: The Web-of-Science annual distribution of Vision Transformer techniques. Since 2020, the number of studies on Vision Transformer applications in WOS has expanded considerably.

 Comment 2:   Figure 6 is important. It should be better described in the caption and contextualized in the text.

Author's response: We thank the reviewer for this comment. We have made the necessary changes to improve the results of our work.

Author's actions: Figure 6, which is now figure 8, is referenced in the figure's title as well as its descriptive text. This image's explanation is also referenced in its caption, where figure 8.a depicts SSA and figure 8.b depicts TSA. Detailed explanations of these concepts are provided along the same lines.

Comment 3: In the experimental part, measures more sensitive to data imbalance should be adopted, such as Matthew’s correlation coefficient or balanced accuracy.

Author's response: We thank the reviewer for this comment. We have made the necessary changes to improve the results of our work.

Author's actions:

The comparison study section was based on two modalities: RGB and Skeleton. The datasets that are used in both of these modalities are the most notable ones where classes are given equal weight, as stated in the “Modality comparative literature subsection”, Section 7,  in RGB and Skeleton. We didn't mention Matthew's correlation or balanced accuracy for that reason.

Comment 4: The datasets used should be described perhaps in tabular form.

Author's response: We thank the reviewer for this comment. We have made the necessary changes to improve the results of our work.

Author's actions:  Considering that we've previously supplied two tabular formats that detail the modalities and their outcomes in various datasets. The datasets that have been recently contributed to this publication are described and added to the revised version of the manuscript.

Comment 5: With reference to figure 6, what happens if we try to reduce the dimensionality of space?

Author's response: We thank the reviewer for this comment. We have made the necessary changes to improve the results of our work.

Author's actions:  Based on the approach of the paper mentioned in Figure 6, which is now Figure 8, (a) represents the spatial representation, while (b) represents the temporal representation. For the spatial representation, the spatial size is fixed. For each frame, each skeleton has 25 joints of 3D coordinates (x, y, z), so the size is 25*3=75 when two skeletons exist in an image, the size is then 75*2=150. For the temporal representation, the segment is thus fixed according to the number of frames.

Comment 6:  In the specific context, the following paper should be cited:

Author's response: We thank the reviewer for this comment. We have made the necessary changes to improve the results of our work

Author's actions: The proposed paper “Manipur, I., Manzo, M., Granata, I., Giordano, M., Maddalena, L., & Guarracino, M. R. (2021). Netpro2vec: a graph embedding framework for biomedical applications. IEEE/ACM Transactions on Computational Biology and Bioinformatics, 19(2), 729-740. “ is added in the revised version of the paper (please see the reference number [83]).

Furthermore, our review of the bibliography has been improved thanks to the addition of fresh references (118 references). Due to the novelty of the subject matter, we have concentrated only on more recent publications (mostly from 2021 and 2022).

Round 2

Reviewer 2 Report

I think the manuscript can be accepted now.

Reviewer 3 Report

The authors have addressed my comments from the previous round. Thus I would suggest acceptance of the manuscript.

Reviewer 4 Report

As far as I'm concerned, there are no further changes to be made